# A direct experimental test of Ohno's hypothesis

Ljiljana Mihajlovic[1], Bharat Ravi Iyengar[2,3], Florian Baier[1], Içvara Barbier[1], Justyna Iwaszkiewicz[4], Vincent Zoete[4,5], Andreas Wagner[2,6,7]*, Yolanda Schaerli[1]*

[1]Department of Fundamental Microbiology, University of Lausanne, Lausanne, Switzerland; [2]Department of Evolutionary Biology and Environmental Studies, University of Zurich, Zurich, Switzerland; [3]Institute for Evolution and Biodiversity, University of Münster, Münster, Germany; [4]Molecular Modeling Group, Swiss Institute of Bioinformatics, Lausanne, Switzerland; [5]Department of Oncology UNIL-CHUV, Ludwig Institute for Cancer Research, University of Lausanne, Epalinges, Switzerland; [6]The Swiss Institute of Bioinformatics, Lausanne, Switzerland; [7]The Santa Fe Institute, Santa Fe, United States

*For correspondence:
andreas.wagner@ieu.uzh.ch (AW);
yolanda.schaerli@unil.ch (YS)

Competing interest: The authors declare that no competing interests exist.

## eLife Assessment

This **fundamental** study uses a creative experimental system to directly test Ohno's hypothesis, which describes how and why new genes might evolve by duplication of existing ones. In agreement with existing criticism of Ohno's original idea, the authors present **compelling** evidence that having two gene copies does not speed up the evolution of a new function as posited by Ohno, but instead leads to the rapid inactivation of one of the copies through the accumulation of mostly deleterious mutations. These findings will be of broad interest to evolutionary biologists and geneticists.

**Abstract** Gene duplication drives evolution by providing raw material for proteins with novel functions. An influential hypothesis by Ohno (1970) posits that gene duplication helps genes tolerate new mutations and thus facilitates the evolution of new phenotypes. Competing hypotheses argue that deleterious mutations will usually inactivate gene duplicates too rapidly for Ohno's hypothesis to work. We experimentally tested Ohno's hypothesis by evolving one or exactly two copies of a gene encoding a fluorescent protein in *Escherichia coli* through several rounds of mutation and selection. We analyzed the genotypic and phenotypic evolutionary dynamics of the evolving populations through high-throughput DNA sequencing, biochemical assays, and engineering of selected variants. In support of Ohno's hypothesis, populations carrying two gene copies displayed higher mutational robustness than those carrying a single gene copy. Consequently, the double-copy populations experienced relaxed purifying selection, evolved higher phenotypic and genetic diversity, carried more mutations and accumulated combinations of key beneficial mutations earlier. However, their phenotypic evolution was not accelerated, possibly because one gene copy rapidly became inactivated by deleterious mutations. Our work provides an experimental platform to test models of evolution by gene duplication, and it supports alternatives to Ohno's hypothesis that point to the importance of gene dosage.

## Introduction

Because gene duplication is frequent in genome evolution (*Lynch and Conery, 2000a*), it is a main source of material for novel proteins. The gene copies resulting from a gene duplication may have a variety of fates. First, loss-of-function mutations may inactivate one of the gene copies or lead to its

**eLife digest** Inside all living things, genes carry instructions to make and maintain the body. Individuals carefully maintain their set of genes, known as the genome, to pass their appearance and other traits on to the next generation. Sometimes a particular gene may be duplicated so that cells end up with an extra copy in their genome.

Typically, around 50% of genes are duplicated in genomes and these duplicates may then accumulate changes (or "mutations") that enable them to adopt new roles in the body. Some mutations may be harmful to the body and lead to the mutated gene being inactivated.

In 1970, the researcher Susumu Ohno proposed that when two copies of the same gene are present, they can accumulate more mutations than a single copy would while maintaining function. As a result, duplicated genes may thus evolve new properties and roles in the body more rapidly than single-copy genes. However, it has been difficult to design experiments to test this hypothesis.

A protein known as GFP emits green light when it absorbs certain colors of light, a phenomenon known as fluorescence. To test Ohno's hypothesis, Mihajlovic et al. developed an experimental system to place one or two copies of the gene that encodes GFP into bacteria known as *Escherichia coli*. The team then introduced mutations into these genes and simulated evolution by selecting bacteria on their ability to emit green, blue or both lights.

The experiments found that part of Ohno's hypothesis is correct. Bacteria with two gene copies were more likely to retain their ability to emit green light after mutations than bacteria with one copy. However, gene duplication did not accelerate the evolution of more fluorescent GFPs or of new functions of the protein, such as emitting blue light. Instead, one copy of the gene often became inactivated by harmful mutations. This suggests that there may be other reasons beyond those proposed by Ohno to explain why gene duplications are so common in nature – for instance, by enhancing protein production.

The experimental system developed in this work serves as a platform for further investigations into Ohno's hypothesis, which may help us better understand the origins of genetic diversity in bacteria and other forms of life.

loss (*Lynch and Conery, 2000a*). Second, both copies might be conserved by natural selection for increased gene dosage (*Kondrashov, 2010*). Third, mutations can cause a partial loss-of-function in either duplicate, such that the pre-duplication functions become partitioned among the duplicates in a process called sub-functionalization (*Force et al., 1999*; *Hittinger and Carroll, 2007*). Finally, duplicates can gradually diverge and evolve distinct novel functions (neo-functionalization) (*Ohno, 1970*).

The factors that determine which of these outcomes occurs are still poorly understood, even though gene duplication has been the subject of many studies (*Force et al., 1999*; *Hittinger and Carroll, 2007*; *Dhar et al., 2014*; *Lynch and Force, 2000b*; *Näsvall et al., 2012*; *Ogino et al., 2016*; *Zhang et al., 2002*; *Voordeckers et al., 2012*; *Pougach et al., 2014*). The last outcome, in which natural selection helps to create functional diversity from complete redundancy, has attracted the most interest, which led to multiple evolutionary models (*Hittinger and Carroll, 2007*; *Ohno, 1970*; *Näsvall et al., 2012*; *Bergthorsson et al., 2007*; *Des Marais and Rausher, 2008*; *Hughes, 1994*). The earliest and most influential was developed by Susumu Ohno in his book '*Evolution by gene duplication*' (*Ohno, 1970*). Ohno wrote that the most important role of a new gene copy is to provide redundancy. The redundant copy could escape from 'the relentless pressure of natural selection' and accumulate 'formerly forbidden mutations' to 'emerge as a new gene locus with a hitherto unknown function' (*Ohno, 1970*). Expressed in modern language, gene duplication increases the mutational robustness of the phenotype produced by the gene duplicates (*Wagner, 2008*), which relaxes the selection pressure on either duplicate, accelerates the accumulation of genetic diversity, and thus facilitates the evolution of new gene functions and phenotypes.

Ohno's model faces a fundamental challenge. Beneficial mutations that create new gene functions are much rarer than deleterious mutations that impair or destroy gene functions. Thus, deleterious mutations may usually cause a loss of function of one duplicate long before rare beneficial mutations lead to its functional divergence. This problem is known as Ohno's dilemma (*Bergthorsson et al., 2007*). It has led to several alternative hypotheses for the emergence of novel protein functions after

gene duplication (reviewed in *Innan and Kondrashov, 2010*). They include the 'mutations during non-functionality' model (MDN) *Hughes, 1994*; the 'Duplication-Degeneration-Complementation' model (DDC) (*Force et al., 1999*), the 'Escape from Adaptive Conflict' model (EAC) (*Hittinger and Carroll, 2007*; *Des Marais and Rausher, 2008*), and the 'Innovation-Amplification-Divergence' model (IAD) (*Näsvall et al., 2012*; *Bergthorsson et al., 2007*). In the last model, a duplicate gene can be temporarily present in more than two copies.

Most empirical data about the evolution of duplicated genes comes from comparative genomic data analyses (*Conant and Wolfe, 2008*). Typically, pertinent studies compare duplicated genes (paralogs) with their single-copy relatives (orthologs) in related species. Quantities like the number of amino acid changes and the ratio between non-synonymous and synonymous mutations are used as proxies for evolutionary rates and selection pressures, respectively. Such studies usually do not provide a phenotypic characterization of duplicates, especially when conducted on a genome-wide scale. Some individual duplicates and their evolutionary trajectories have been studied in more detail (*Ogino et al., 2016*; *Zhang et al., 2002*; *Voordeckers et al., 2012*; *Pougach et al., 2014*). However, the comparative method mainly captures duplicates that are already fixed, functional, and selectively maintained (*Force et al., 1999*; *Hittinger and Carroll, 2007*; *Lynch and Force, 2000b*). In addition, little information is usually available about the environmental conditions under which these duplicated genes have evolved. Moreover, the comparative method cannot easily distinguish the contribution of mutations to the divergence of a gene's coding region, to its regulation *Hittinger and Carroll, 2007*, and to its interaction with other genes and their products (e.g. in protein complexes or as part of a regulatory network) (*Pougach et al., 2014*; *Teichmann and Babu, 2004*). All of these factors can be important for the evolutionary fate of duplicated genes.

Experimental evolution (*Arnold, 2018*; *Elena and Lenski, 2003*) can mitigate the shortcomings of the comparative method. The DNA of populations evolved in the laboratory can be sequenced during any such experiment to reconstruct their evolutionary dynamics in detail. Any phenotype of interest can be carefully monitored during evolution. Thus, experimental evolution can help study both genotypic and phenotypic evolution. In addition, evolution experiments can be designed to test specific hypotheses, such as Ohno's hypothesis.

A type of evolution experiment well-suited to testing Ohno's hypothesis is directed evolution, which evolves individual genes by repeated rounds of mutagenesis and selection for specific phenotypes (*Arnold, 2018*). Directed evolution is not only highly useful for generating commercially important proteins, it can also provide fundamental insights about protein evolution (*Soskine and Tawfik, 2010*). Fluorescent proteins are especially useful for such experiments, because their phenotype can be measured in every individual of large microbial populations (*Iyengar and Wagner, 2022a*; *Zheng et al., 2020*). However, few studies have used experimental evolution thus far to investigate the evolution of duplicated genes. These studies underline the importance of gene duplication in providing fast adaptation under changing environmental conditions. In some studies one copy was lost (*Dhar et al., 2014*; *Holloway et al., 2007*), while in others, additional copies were gained (*Näsvall et al., 2012*; *Kugelberg et al., 2006*; *Tomanek et al., 2020*). Together these studies highlight that gene dosage and selection for dosage can play an important role during the evolution of duplicated genes (*Dhar et al., 2014*; *Näsvall et al., 2012*; *Holloway et al., 2007*; *Kugelberg et al., 2006*; *Tomanek et al., 2020*; *Tomanek and Guet, 2022*).

These studies also raise the question of whether gene duplication can provide an advantage beyond its effects on gene dosage. To find out it is necessary to study the evolution of gene duplicates while keeping the copy number of the duplicated gene exactly at two. This is challenging because gene duplication causes recombinational instability and high variability in copy number. No previous experimental studies were designed to control copy number. Here, we present an experimental system that allowed us to keep the copy number fixed at one or two genes, and to follow the evolution of each gene copy in the absence of any dosage increase.

Briefly, we expressed a fluorescent protein in one or two copies in *Escherichia coli (E. coli)* and subjected it to multiple rounds of directed evolution under different conditions. We then analyzed the evolving populations at both the genotypic and phenotypic levels. We observed that in the early stages of evolution, populations carrying two gene copies displayed higher mutational robustness compared to those carrying a single gene copy. As a consequence, populations with two copies experienced relaxed purifying selection, evolved higher phenotypic and genetic diversity, carried more

mutations, and accumulated combinations of key mutations earlier than the control populations with one copy. However, this did not manifest in accelerated phenotypic evolution in double-copy populations, possibly because one gene copy rapidly becomes inactivated by deleterious mutations.

## Results

### Experimental design

To experimentally study the evolutionary consequences of gene duplication, we performed directed evolution of a fluorescent protein, that is expressed in *E. coli* from a single gene (single copy) or two identical genes (double copy). Our study protein was *coGFP*, a green fluorescent protein first isolated from the marine cnidarian *Cavernularia obesa* (*Ogoh et al., 2013*). Our choice was motivated by the fact that this protein exhibits a dual phenotype that let us explore multiple selection regimes. When excited with light at a wavelength of 388 nm, its emission spectrum has two maxima, the first corresponding to blue color (456 nm) and the second corresponding to green color (507 nm), which may be caused by two different protonation states of the chromophore (*Ogoh et al., 2013*; *Hanson et al., 2002*). Because the ratio of blue and green emission is pH sensitive, we used buffers to maintain a constant pH during evolution (Methods). Therefore, any changes in fluorescence we observed during evolution are due to genotypic changes caused by mutations. Amino acid substitutions at position 147 of this GFP lead to different ratios of blue and green fluorescence (*Ogoh et al., 2013*). The starting point of our evolutionary experiments was a variant with a glycine at residue 147 that fluoresces about

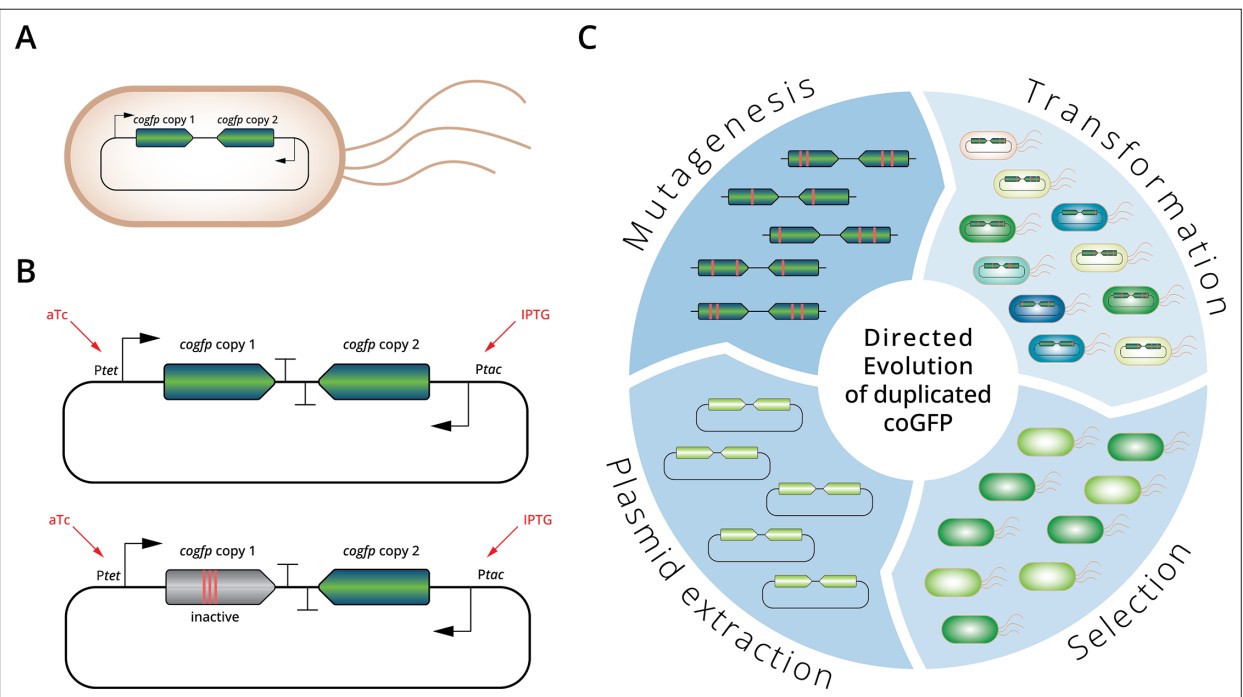

**Figure 1.** Experimental evolution of a duplicated fluorescent protein. (**A**) *E. coli* cells carry a plasmid containing a duplicated gene coding for coGFP. (**B**) Upper panel: Plasmid with two copies of the cogfp gene, both under control of independently inducible promoters (P$_{tet}$ and P$_{tac}$); lower panel: control single-copy plasmid with only one active gene copy, the other copy is not fluorescent due to mutations engineered into the chromophore. (**C**) Overview of the directed evolution experiment of the duplicated cogfp gene (see text for details).

The online version of this article includes the following figure supplement(s) for figure 1:

**Figure supplement 1.** Dual-color-emitting fluorescent protein coGFP.

**Figure supplement 2.** Plasmid map of pDUP2 carrying the duplicated co*gfp* gene.

**Figure supplement 3.** Expression levels of single-copy and double-copy constructs.

**Figure supplement 4.** Selection regimes based on cell's fluorescence phenotypes.

**Figure supplement 5.** Applied selection stringencies.

equally brightly in green and blue (*Figure 1—figure supplement 1*). We call this variant our ancestral protein.

We mimicked the duplication event by placing two copies of the *cogfp* gene on a plasmid (*Figure 1*, *Figure 1—figure supplement 2*). To avoid recombinational copy number instability (*Dhar et al., 2014*), we inserted the two copies in opposing (convergently transcribed) directions. We placed the two copies under independent control of the inducible promoters $P_{tet}$ (inducible by anhydrotetracycline (aTc)) and $P_{tac}$ (inducible by isopropyl-β-D-1-thiogalactopyranosid (IPTG)), which allowed us to express either both copies or only one copy at a time. To compare the evolution of a duplicated gene to the evolution of a single gene, we created an identical control plasmid with two gene copies, but inactivated one of the copies by three amino acids substitutions in the chromophore region (Q74A, Y75S, G76A) (*Barondeau et al., 2005*). The advantage of this control single-copy plasmid is that it is the same size as the double-copy plasmid with two intact copies, and produces the same amount of protein, which allowed us to avoid differences in plasmid size and protein expression as confounding factors for our experiments. When induced with both aTc and IPTG the double-copy plasmid shows ~1.5-fold higher fluorescence than the single-copy plasmid (*Figure 1—figure supplement 3*).

We performed directed evolution of both single- and double-copy populations by repeated rounds ('generations') of mutagenesis, cloning of mutagenized *cogfp* genes into fresh (unmutated) plasmid backbone, transformation of *E. coli* cells with the resulting plasmid libraries, and selection using fluorescence-activated cell sorting (FACS). After each selection step, we isolated plasmids from the cells that had survived selection to start the next generation (*Figure 1*). Population (library) sizes exceeded $10^4$ throughout our experiments, rendering the effects of genetic drift negligible on the time scale of our experiment.

Because the opposing directions of the two gene copies renders mutagenesis by the frequently used error-prone polymerase chain reaction impossible (*Baier et al., 2019*; *Lagator et al., 2017*; *Wolf et al., 2015*), we mutagenized our genes through isothermal DNA amplification (*Dean et al., 2001*; *Lizardi et al., 1998*). This method introduced approximately two amino acid substitutions into the coding region of each *cogfp* gene per generation of mutagenesis (*Source data 1*). After mutagenesis, we isolated the two *cogfp* coding sequences together before inserting them into a fresh plasmid backbone. Because this re-insertion could occur in both directions, a gene that was controlled by the $P_{tet}$ promoter in one generation might come under the control of $P_{tac}$ in the next generation, and vice versa.

Applying selection on fluorescence via FACS allowed us to apply multiple selection regimes. Specifically, our experiments used the following selection regimes: *green*: selection for green (no selection against blue fluorescence); *blue*: selection for blue (no selection against green fluorescence); *green and blue:* selection for both green and blue; *green-only*: selection for green and against blue; *blue-only*: selection for blue and against green; and *no selection* for either fluorescence color (*Figure 1—figure supplement 4*).

For each of these regimes (except the no selection regime), we increased the strength of selection during our experiment. Specifically, we only applied weak selection to accumulate genetic diversity in the first generation, allowing 60% of cells displaying the highest fluorescence to survive. In generation two, we allowed the top 1% of cells to survive, and in generations 3–5, we allowed only the top 0.01% of cells to survive (*Figure 1—figure supplement 5*). We performed triplicate experiments for each of the above six selection regimes, and for each of the single-copy and double-copy plasmids, leading to a total of 3×6×2=36 evolution experiments, each of which lasted five generations.

## Gene duplication increases mutational robustness

One central prediction of Ohno's hypothesis is that gene duplication increases the mutational robustness of gene duplicates (*Ohno, 1970*). In our study system, robustness refers to the preservation of the fluorescence phenotype after DNA mutagenesis (*Figure 2—figure supplement 1*). To test this prediction, we determined the fraction of cells that were still able to maintain their fluorescence after mutagenesis (shown for *green* selection regime in *Figure 2*, and for the other regimes in *Figure 2—figure supplement 2*). In the first generation of the *green* selection regime, mutagenesis (before applying any selection) led to loss of fluorescence in 73.7% of variants from single-copy plasmids, but only in 59% of variants from double-copy plasmids. Thus, our gene duplicates are indeed more robust to mutations. This difference can be explained with a simple probability calculation, based on

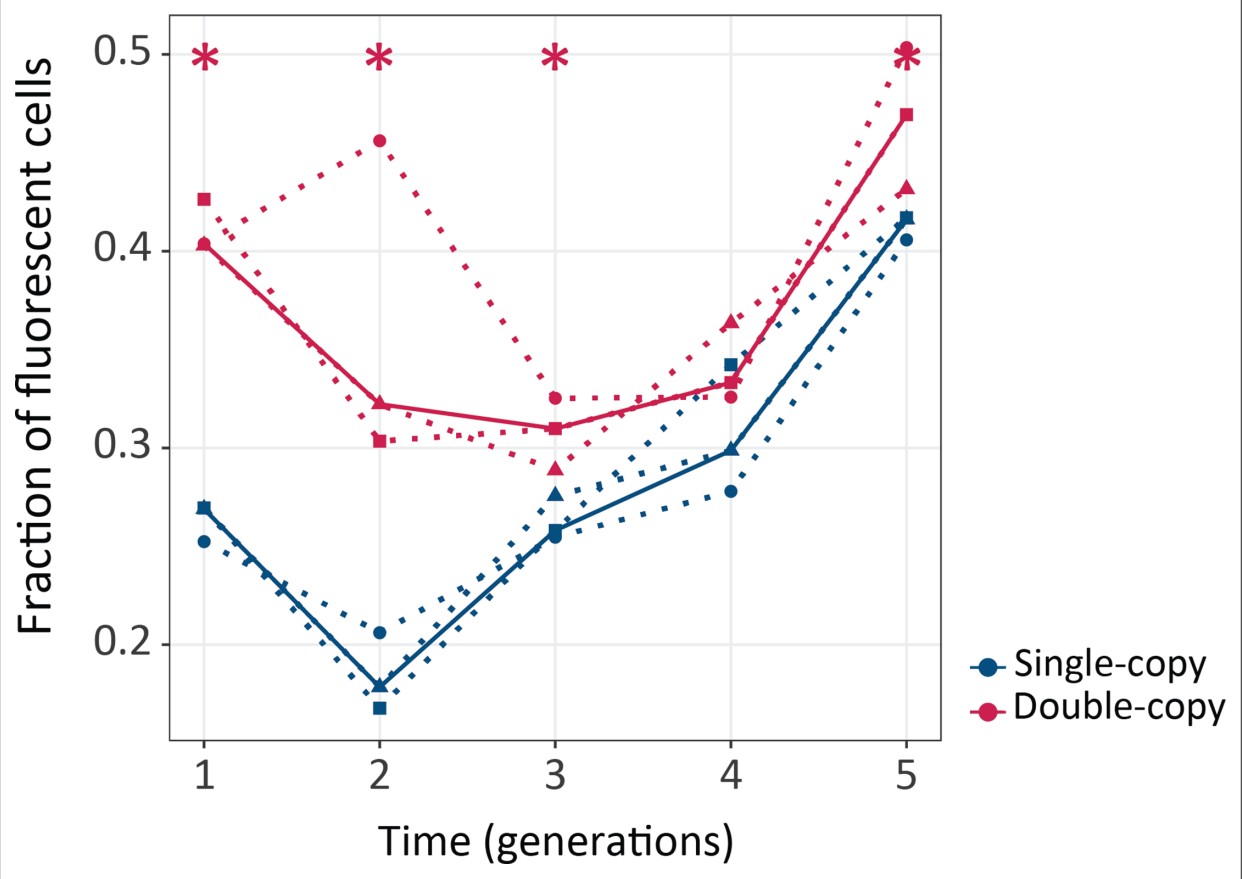

**Figure 2.** Gene duplication increases mutational robustness. The vertical axis shows mutational robustness, measured as the percentage of cells that maintain their fluorescence after mutagenesis, as a function of time (in generations of directed evolution) on the horizontal axis. Thick blue and red lines stand for the median fraction of fluorescent cells for single-copy and double-copy mutant libraries, respectively, while dotted lines indicate data from the three biological replicates. One-tailed Mann-Whitney tests, *p≤0.05, n=3. (*Figure 2—source data 1*).

The online version of this article includes the following source data and figure supplement(s) for figure 2:

**Source data 1.** Data plotted in *Figure 2*.

**Figure supplement 1.** Quantification of fluorescent cells.

**Figure supplement 2.** Gene duplication increases mutational robustness.

our observation that mutagenesis preserves fluorescence in a single-copy of coGFP with a probability of 0.26. Specifically, this calculation predicts that $100 \times 0.74^2 = 55\%$ of members of the double-copy population have no active copy anymore, which is close to the 60% we observed.

In the second generation of evolution, the cumulative effects of mutations caused a greater fraction of variants to lose fluorescence after mutagenesis, but here too double-copies remained more robust to mutations (with 36% of fluorescing cells after mutagenesis) than single-copies (18.4% of fluorescing cells; *Figure 2*). From the third generation onward, robustness increases steadily in both kinds of populations (*Figure 2*). We also observed the same patterns of evolution – greater robustness of double-copy populations early on, increased robustness in all populations – also occur for the other selection regimes (*Figure 2—figure supplement 2*). An exception is the *no selection* regime, where robustness does not increase (*Figure 2—figure supplement 2*). This observation demonstrates that selection is essential to increase mutational robustness in our experiment. Altogether these data show that gene duplication indeed increases mutational robustness.

Next, we analyzed the fluorescence phenotype of evolved populations (*Figure 3—figure supplement 1*). We found that populations subject to selection for green fluorescence changed their phenotype more dramatically than for other selection regimes. To keep this manuscript short, we thus

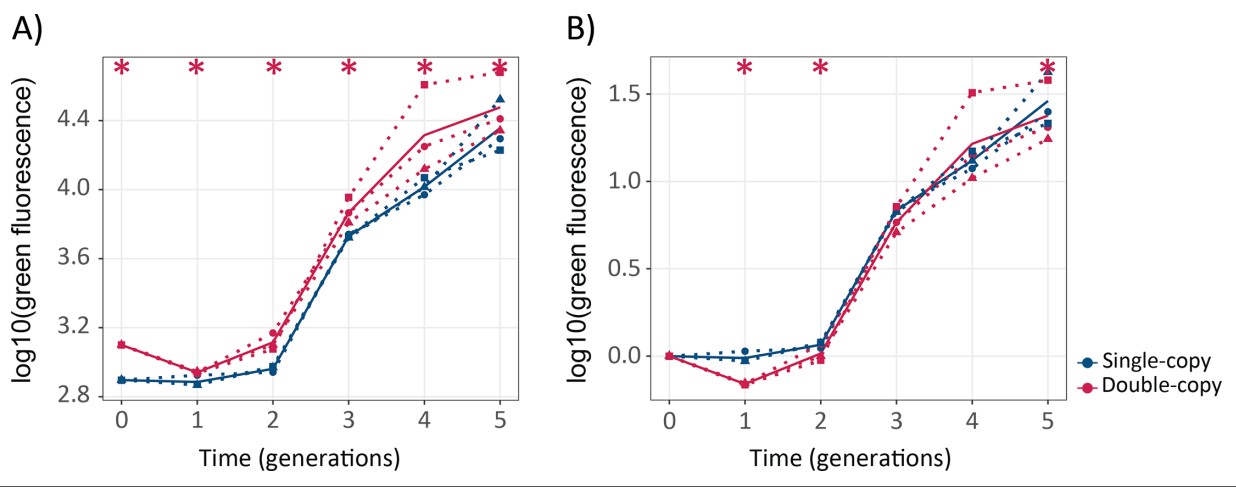

**Figure 3.** Gene duplication does not lead to significantly faster evolution of green fluorescence. (**A**) The vertical axis shows green fluorescence (log10) as a function of time (generations) on the horizontal axis. (**B**) The vertical axis shows green fluorescence (log10) normalized to the fluorescence of its ancestral population as a function of time (generations) on the horizontal axis. Thick blue and red lines stand for the median fluorescence of single-copy and double-copy mutant populations, respectively, while dotted lines indicate data from the three biological replicates. One-tailed Mann-Whitney tests, *p≤0.05, n=3. (*Figure 3—source data 1*).

The online version of this article includes the following source data and figure supplement(s) for figure 3:

**Source data 1.** Data plotted in *Figure 3*.

**Figure supplement 1.** Fluorescence levels during evolution experiment.

**Figure supplement 2.** Coefficient of variation of fluorescence.

**Figure supplement 3.** Copy maintenance vs copy loss.

focus on the *green* regime in the main text and relegate data for the other selection regimes to the supporting information.

## Gene duplication does not accelerate phenotypic evolution significantly, but increases phenotypic diversity

Another key prediction of Ohno's hypothesis is that gene duplication facilitates phenotypic evolution (*Ohno, 1970*). To test this prediction, we asked whether green fluorescence evolves faster in double-copy than in single-copy populations (*Figure 3*). Double-copy populations evolved under the green selection regime displayed statistically significantly higher fluorescence than the single-copy populations throughout the five rounds of evolution (*Figure 3A*; one-tailed Mann-Whitney p≤0.05, n=3). However, if we normalize the fluorescence of the evolving populations to the fluorescence of the corresponding ancestral single- or double-copy populations (*Figure 3B*), it is the single-copy populations that show a statistically significant higher fluorescence in generations 1, 2, and 5 (one-tailed Mann-Whitney p≤0.05, n=3), whereas in rounds 3 and 4 no significant difference between single-copy and double-copy populations exists. In other words, the double-copy population does not lead to faster evolution of green fluorescence beyond the increased gene dosage provided by the second copy. However, we observed that the variance in fluorescence within a population is significantly greater in double-copy populations than in single-copy populations (*Figure 3—figure supplement 2*).

## One gene copy rapidly loses function after duplication

According to Ohno, a duplicated gene copy is not subject to selection for its initial function, and can thus accumulate previously 'forbidden mutations' that might eventually lead to a new phenotype (*Ohno, 1970*). However, both original and copy may also suffer a loss of function through deleterious mutations (*Bergthorsson et al., 2007*) that may, for example, prevent proper folding of the encoded protein. Our experimental design allowed us to induce the two *cogfp* copies in a double-copy population independently from each other, which can help us to find out whether one or both copies are active in cells of a double-copy population. To this end, we chose a random sample of ~300 *cogfp*

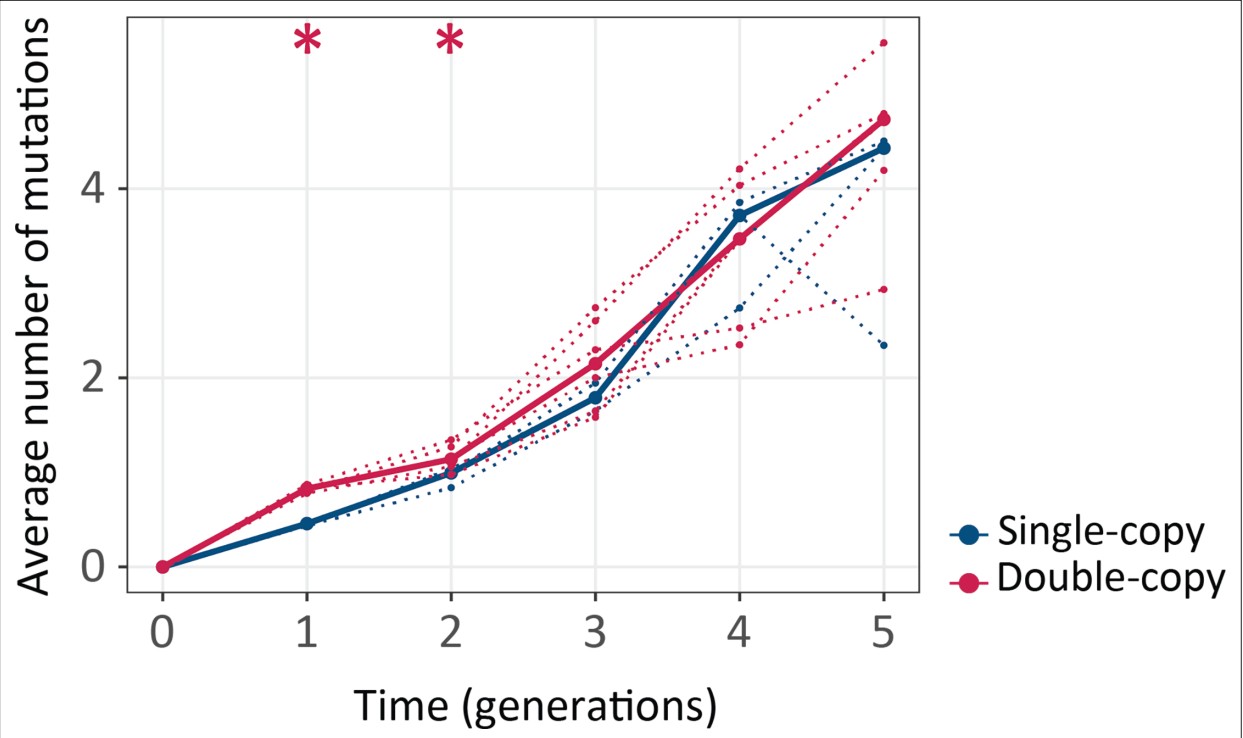

**Figure 4.** Double-copy populations accumulate more mutations per gene than single-copy populations during the first two generations. The vertical axis shows the average number of non-synonymous mutations per cogfp gene, as a function of time (in generations of directed evolution) on the horizontal axis. Thick blue and red lines stand for the median numbers of single-copy and double-copy mutant populations respectively, while dotted lines indicate data from the three biological replicates. Mann-Whitney tests *p≤0.05, n=3. (*Figure 4—source data 1*).

The online version of this article includes the following source data and figure supplement(s) for figure 4:

**Source data 1.** Data plotted in *Figure 4*.

**Figure supplement 1.** Average number of mutations for single- and double-copy populations evolved under the indicated selection regimes.

**Figure supplement 2.** Frequency distribution of the number of mutations.

variants from the double-copy populations for each generation and measured their fluorescence either under the induction of copy 1 (with aTc), of copy 2 (with IPTG), of both copies (with aTc and IPTG), and under no induction (*Figure 3—figure supplement 3*). The inactivation of one copy was indicated by the absence of fluorescence in the presence of one inducer. We found that the percentage of variants with two active copies decreased dramatically over time. Already after one generation of evolution, 38% of cells in a population harbored an inactive *cogfp* copy. This incidence of inactive copies subsequently increased further to 81% after generation four.

## Populations with two gene copies carry more mutations per gene early during post-duplication evolution

After these phenotypic analyses, we studied the evolutionary dynamics of genetic variation in more detail with high-throughput single-molecule real-time (SMRT) sequencing (*Wenger et al., 2019*). This long-read sequencing method allowed us to genotype an entire 1538 bp double-copy sequence variant in a single high-quality sequence read, thus preserving the linkage between two gene copies that evolved together. After quality filtering (Methods), we were left with 279–36,000 variants per population. We focused our analysis on non-synonymous mutations, and first studied the frequency distribution of mutations. We found that *cogfp* genes in double-copy populations harbor more mutations than in single-copy populations during the first two rounds (*Figure 4* and *Figure 4—figure supplement 1*, *Figure 4—figure supplement 2*). Specifically, the mean number of mutations per *cogfp* gene in double-copy/single-copy populations for generations 1–5 during selection for green

fluorescence was 0.827/0.455, 1.155/0.956, 2.144/1.794, 3.34/3.436, 4.488/3.758 and the differences are significant (Mann-Whitney test, p≤0.05) for the first two rounds.

This finding is consistent with the increased mutational robustness provided by gene duplication in the early rounds of evolution (*Figure 2*). A similar pattern exists for the other selection regimes, with even bigger differences between double- and single-copy populations during selection for blue fluorescence (*Figure 4—figure supplement 1*, *Figure 4—figure supplement 2*).

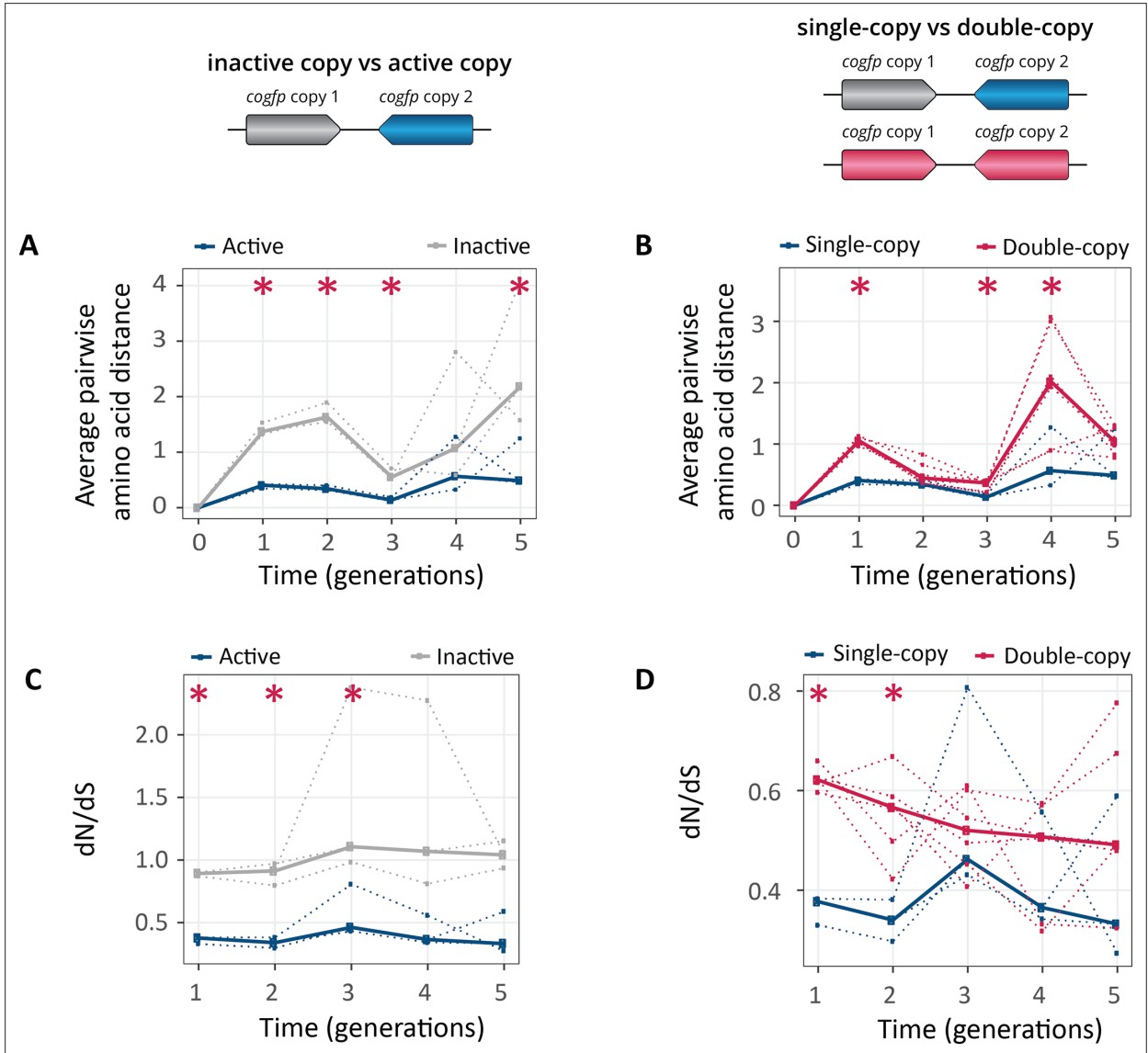

**Figure 5.** Populations with two gene copies show increased genetic diversity and dN/dS ratios. The horizontal axes of all panels show time in generations during selection for green fluorescence. (**A**) Average pairwise amino acid distance for coGFP molecules in single-copy populations (blue: active copy, gray: inactive copy), (**B**) average pairwise amino acid distance for coGFP molecules of the active copy in single-copy populations (blue) vs both copies in double-copy populations (pink). (**C**) dN/dS ratio in single-copy populations (blue: active copy, gray: inactive copy), (**D**) dN/dS ratio of the active copy in single-copy populations (blue) vs both copies in double-copy populations (pink). Thick lines represent the median over three replicate populations, while dotted lines indicate data from the individual biological replicates. *p≤0.05 Mann-Whitney test, n=3. (*Figure 5—source data 1*).

The online version of this article includes the following source data and figure supplement(s) for figure 5:

**Source data 1.** Data plotted in *Figure 5*.

**Figure supplement 1.** Populations with two gene copies are showing increased genetic diversity.

**Figure supplement 2.** Populations with two gene copies are showing higher dN/dS ratios.

## Gene duplication increases genetic diversity

The higher number of mutations in double-copy populations suggests that *cogfp* genes in such populations diversify more rapidly. To find out whether this is the case, we calculated the average pairwise distance between genotypes within a population, i.e., the average number of amino acid differences between any two *cogfp* variant sequences during selection for green fluorescence (*Figure 5A–B*, *Figure 5—figure supplement 1*). Recall that even the single copy populations carry actually two *cogfp* genes, but one copy is inactivated. This inactive copy gives us a baseline of how much genetic diversity a neutrally evolving gene acquires. Therefore, we first compared the active versus the inactive copies in the single-copy populations. As expected, in single-copy populations, the active *cogfp* copy diversified less rapidly than the inactive copy (*Figure 5A*), indicating that selection eliminated some mutations in the active copy. Next, we compared the active copy of single-copy populations with the double-copy populations. We found that indeed individual *cogfp* copies in double-copy populations diversified more rapidly than in single-copy populations with a significant difference in generations 1, 3, and 4 (*Figure 5B* and $p \leq 0.05$ Mann-Whitney test, n=3). The increased genetic diversity is consistent with the increased phenotypic diversity (*Figure 3—figure supplement 2*), the increase in mutational robustness (*Figure 2*), and the rapid inactivation of one of the duplicated copies (*Figure 3—figure supplement 3*).

Complementary evidence for differences in genetic diversity comes from the ratio between the number of non-synonymous mutations per non-synonymous site (dN) to the number of synonymous mutations per synonymous site (dS) (*Figure 5C–D* and *Figure 5—figure supplement 2*). This dN/dS ratio is commonly used to identify the kind of selection acting on evolving genes (*Goldman and Yang, 1994*; *Kimura, 1977*; *Kondrashov et al., 2002*; *Kryazhimskiy and Plotkin, 2008*; *Moore and Purugganan, 2003*): dN/dS >1 indicating positive selection, dN/dS = 1 indicating neutral evolution and dN/dS <1 indicating negative (purifying) selection. The inactive copy in single-copy populations has a dN/dS≈1 (*Figure 5C*), as expected for a copy evolving neutrally. In double-copy populations, dN/dS <1 averaged over both copies is smaller than one, but significantly greater than for the active copy in single-copy populations in generations 1 and 2 (*Figure 5D*). This shows that despite selection for increased fluorescence, most mutations experienced by both single-copy and double-copy *cogfp* genes are deleterious and subject to purifying selection. In other words, mutations that increase fluorescence are in the minority. Purifying selection is weaker in double-copy than in single-copy populations (*Figure 5D*), a pattern that is again consistent with an increase in mutational robustness (*Figure 2*) and rapid inactivation of one copy in at least some individuals of double-copy populations (*Figure 3—figure supplement 3*). The rapid inactivation might also lead to re-intensified purifying selection on the remaining active copy. This might explain why in later generations the differences between single-copy and double-copy populations are not statistically significant anymore. Analogous patterns hold for the other selection regimes we studied (*Figure 5—figure supplement 2*).

## G147S, V162D, and L98M rise to high frequencies

In our next analysis, we identified the mutations that attained the highest population frequencies and are thus probably responsible for the changes in fluorescence during evolution (*Figure 6—figure supplement 1*). The mutations G147S and V162D swept to the highest frequency under selection for green fluorescence. While mutations at position 147 have been previously described to influence coGFPS's fluorescence (*Ogoh et al., 2013*), the mutation with the highest frequency V162D, which reaches a median frequency of 99% in both single- and double-copy populations, has not been described as such. Both G147S and V162D lead to a more polar environment around chromophore residue Y75, thus facilitating proton transfer and green light emission (*Hanson et al., 2002*). One additional mutation (L98M) reached high frequency under selection for green fluorescence, but it also did so in all other selection regimes (*Figure 6—figure supplement 1*). Like L98M, the additional mutations S2I, V141I, and V25L also occurred in all selection regimes, but they reached lower frequencies than L98M during the five generations of the experiment. We hypothesized that mutations observed in all selection regimes do not derive their benefit from increasing the intensity of any one fluorescent color. Instead, they may increase protein expression, solubility, or thermal stability.

To confirm the phenotypic effects of these mutations, we engineered them individually into the ancestral protein sequence and measured the fluorescence of the resulting protein variants both when expressed in *E. coli* cells and as purified proteins in vitro (*Figure 6—figure supplement 2*,

*Figure 6—figure supplement 3*). This analysis confirmed the green-shifting effect (i.e. an increase in the ratio of green to blue fluorescence) of the mutations G147S and V162D. Interestingly, the purified protein carrying the green-shifting mutation G147S does not fluoresce more intensely than the ancestral protein, but *E. coli* cells expressing this variant do. A Western blot analysis revealed that this mutation increases coGFP expression in *E. coli* cells (*Figure 6—figure supplement 4*). The mutation L98M increases fluorescence at all wavelengths, and it does so by increasing the fraction of folded protein (*Figure 6—figure supplement 5*). In agreement with our experimental observations, computational homology modeling (*Figure 6—figure supplement 6*, *Appendix 1—table 1*) predicts that mutations G147S and L98M increase structural stability.

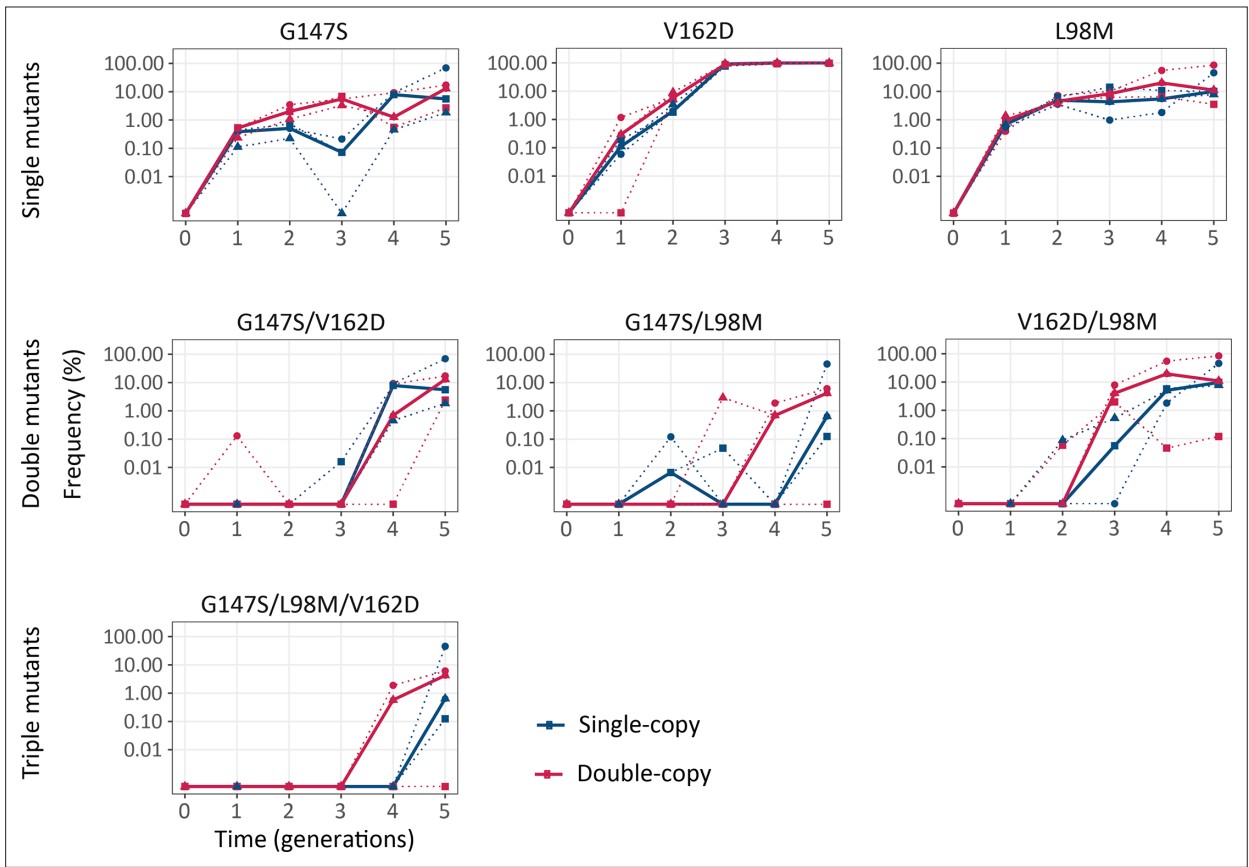

**Figure 6.** Combination of beneficial mutations emerged earlier in double-copy populations. The vertical axis shows the frequency of the indicated mutations and combinations thereof in the populations under selection for green fluorescence, as a function of time (in generations of directed evolution) on the horizontal axis. Thick blue and red lines stand for the median frequencies for single-copy and double-copy populations, respectively, while dotted lines indicate data from the three biological replicates. Detailed statistics are reported in *Appendix 1—table 2*. (*Figure 6—source data 1*).

The online version of this article includes the following source data and figure supplement(s) for figure 6:

**Source data 1.** Data plotted in *Figure 6*.

**Figure supplement 1.** Enriched mutations.

**Figure supplement 2.** Emission spectra of engineered coGFP variants.

**Figure supplement 3.** Relative green fluorescence of the engineered variants compared to the ancestral variant.

**Figure supplement 4.** Expression levels of the engineered coGFP variants.

**Figure supplement 4—source data 1.** Raw and uncropped blots.

**Figure supplement 5.** Analysis of fractions of folded protein.

**Figure supplement 6.** Structure of coGFP.

**Figure supplement 7.** Analysis of variants with two active gene copies.

We also engineered the double (G147S/L98M, V162D/L98M, G147S/V162D) and triple (G147S/V162/L98M) variants and characterized their phenotypes (*Figure 6—figure supplement 3*, *Figure 6—figure supplement 4*). The double and triple mutants fluoresce more brightly in green than any of the single mutants, with the triple mutant G147S/V162/L98M being the brightest (*Figure 6—figure supplement 3*).

## Combination of beneficial mutations emerged earlier in double-copy populations

After identifying and characterizing these key mutations, we asked if they spread more rapidly in double-copy populations than in the single-copy populations (*Figure 6*). Theoretically, the probability that at least one gene copy experiences a specific mutation is two times higher in double-copy populations than in single-copy populations, but subsequently, the frequencies of these mutations are dictated by the strength of selection for them. We found that the double-copy populations have a higher median frequency of mutation G147S in generations 2 and 3, of mutation V162D in generations 1 and 2, and L98M in generations 3 through 5 (Generalized linear model, ANOVA $p < 0.05$, *Appendix 1—table 2*) than single-copy populations (*Figure 6*). The frequencies of combinations of these beneficial mutations were low and variable between replicates, making it difficult to draw strong conclusions about their possible phenotypic effects in the evolving populations. Nevertheless, the multisite mutations G147S/L98M, V162D/L98M, and G147S/V162/L98M emerged earlier in double-copy populations, than in single-copy populations. For example, the median frequency of V162D/L98M exceeded 1% in all three replicates of double-copy populations at generation 3, whereas it exceeded this frequency in single-copy populations only at generation 4. In addition, mutant V162D/L98M was significantly more abundant in double-copy populations from generations 3 through 5 (Generalized linear model, ANOVA $p < 0.05$, *Appendix 1—table 2*, *Figure 6*).

## Discussion

Our aim was to study whether gene duplication can affect mutational robustness and phenotypic evolution beyond any effect of increased gene dosage provided by multiple gene copies. To this end, we needed to maintain a constant and stable copy number of duplicated genes during experimental evolution. This is challenging because of the recombinational instability inherent to repetitive DNA (*Dhar et al., 2014*; *Brown et al., 1998*; *Dunham et al., 2002*). By drawing on our experience of using synthetic biology to study molecular evolution (*Schaerli et al., 2018*; *Baier et al., 2023*; *Santos-Moreno et al., 2023*; *Baier and Schaerli, 2021*), we designed a plasmid system that meets this challenge. It allowed us to evolve a fluorescent protein encoded by either double-copy or single-copy *cogfp* genes.

Using this system, we experimentally confirmed the prediction that gene duplication increases mutational robustness (*Ohno, 1970*) and renders cells more resistant to genetic perturbations (*Payne and Wagner, 2019*; *Wagner, 2005*). Specifically, we showed that mutations are less likely to eliminate fluorescence in evolving populations with two gene copies than with one gene copy (*Figure 2*) during the first two generations of our evolution experiment. Thereafter, the difference in robustness between single-copy and double-copy populations disappeared (*Figure 2*). A likely reason is that in double-copy populations mutations rapidly deactivate one of the two copies (*Figure 3—figure supplement 3*).

It is worth noticing that robustness increased throughout the experiment. Specifically, from the third generation onwards, robustness was always higher than in the preceding generation for the single-copy populations and the same was true for double-copy populations starting in the fourth generation (*Figure 2*). This is likely due to the accumulation of stabilizing mutations. Indeed, we observed that such a mutation, L98M, became fixed in all populations (*Figure 6*). This is in agreement with previous work demonstrating that stabilizing mutations increase robustness and tend to accumulate in directed evolution experiments (*Bratulic et al., 2015*; *Socha and Tokuriki, 2013*; *Tokuriki et al., 2008*).

The increased mutational robustness led to the accumulation of more mutations per *cogfp* genes in double-copy populations (*Figure 4*), and to greater genetic diversity (*Figure 5*) during early post-duplication evolution. The increased robustness also caused a relaxation of selection, manifested in

the ratio dN/dS of non-synonymous to synonymous substitutions (*Figure 5CD*; *Goldman and Yang, 1994*; *Kimura, 1977*). This ratio stayed below one, indicating purifying selection, in both single-copy and double-copy populations, but it was significantly higher in double-copy populations, pointing to relaxed purifying selection in these populations during the first three generations. Our observations are consistent with data from comparative genomics studies, which show relaxed selection shortly after gene duplication (*Lynch and Conery, 2003*). They also show that this relaxation is very short-lived, because most duplicates become either deactivated or diverge in function shortly after duplication.

Despite our directional selection for increased green fluorescence, dN/dS stayed below one throughout the evolution experiment (*Figure 5CD*). This can be explained by the observation that there are far fewer beneficial mutations than deleterious mutations (*Sarkisyan et al., 2016*). Indeed, we identified and characterized only three key mutations that reached high frequencies in all populations selected for green fluorescence (*Figure 6*): G147S, V162D, and L98M. While G147S has previously been described to influence the fluorescence of coGFP (*Ogoh et al., 2013*), to the best of our knowledge we are the first to report the green-shifting effect of V162D in coGFP (*Figure 6—figure supplement 2*). Both mutations are predicted to make the environment around chromophore residue Y75 more polar (*Figure 6—figure supplement 6*), thus promoting proton transfer and green emission (*Hanson et al., 2002*). In contrast, L98M does not change the fluorescence directly, but increases the structural stability of the protein (*Appendix 1—table 1*) and the fraction of folded protein (*Figure 6—figure supplement 5*).

Although our ability to regulate the expression of the two *cogfp* copies independently allows us to study which of the two copies remains active in individual clones of a population, our experimental design does not allow us to distinguish active from inactive copies in a whole population from sequencing data alone. The reason is that we do not know which mutations (and combinations thereof) have deleterious effects, with the exception of (rare) premature stop codons and frame-shift mutations. In consequence, we cannot determine with certainty whether increased mutational robustness or rapid deactivation of one of two duplicates is primarily responsible for the increased accumulation of diversity in double-copy populations. In the future, sorting the double-copy population for variants with one active gene copy and for variants with two active gene copies, followed by high-throughput sequencing of these two subpopulations, might help to quantify the relative contributions of mutational robustness and gene inactivation.

Ohno also predicted that the increased robustness and genetic diversity entailed by gene duplication facilitates phenotypic evolution (*Ohno, 1970*). To validate this prediction, we evolved coGFP populations under directional selection for green fluorescence, and asked whether double-copy populations change their phenotype faster. Green fluorescence was higher in double-copy populations throughout directed evolution, but the difference was smaller than the initial difference caused by the duplication (*Figure 3*). It is worth noting that we evolved each of our single- and double-copy populations separately and in parallel to follow their individual evolutionary trajectories. In a natural population, individuals with one or two copies might occur in the same population and compete against each other. In this situation any dosage advantage of a duplicate gene would itself entail selective benefit. Our approach allowed us to find out if gene duplication facilitates phenotypic evolution beyond any such gene dosage effect. At least for the specific gene, selection pressures, and mutation rates we used, the data suggest that it does not.

Our data also suggests two candidate explanations for this finding. The first is that the genetic changes needed to alter fluorescence in coGFP are simple and mediated by single amino acid changes, such as G147S and V162D. Both changes provide a bigger fitness (fluorescence) advantage to their carrier than the increased gene dosage of double-copy populations (compare *Figure 3*, *Figure 6—figure supplement 2*). Given our high mutation rates and large population sizes (>10,000 individuals) both mutations occur early during evolution and reach a frequency >0.1% in the first generation and in both single-copy and double-copy populations. If multiple mutations were crucial to increase fluorescence, double-copy populations may have a greater advantage. This possibility is consistent with our observation that combinations of beneficial mutations emerged earlier in double-copy populations (*Figure 6*). A second, non-exclusive explanation may be the rapid inactivation of one gene copy by deleterious mutations in double-copy populations, which affected 38% of individuals in generation 1, and up to 81% after generation four (*Figure 3—figure supplement 3*). This means that

any beneficial mutations that convey an advantage to double-copy populations must occur very early after duplication.

It will be important to confirm our findings with other genes and different selection pressures. The reason is that the dynamics of mutation-driven phenotypic evolution after gene duplication may depend on gene-specific details, such as the fraction of mutations that are deleterious, or the number of mutations that are needed for a phenotypic change.

The challenge that a duplicated gene copy must remain free of frequent deleterious mutations long enough to acquire beneficial mutations that provide a new selectable phenotype is known as Ohno's dilemma (*Bergthorsson et al., 2007*). Our experiments confirm that this challenge is highly relevant for post-duplication evolution. Other models such as the innovation-amplification-divergence (IAD) model (*Näsvall et al., 2012*; *Bergthorsson et al., 2007*) postulate that this dilemma can be resolved through an increase in gene dosage that allows latent pre-duplication phenotypes to come under the influence of selection. To distinguish between the effects of gene dosage and other benefits of gene duplication, we prevented recombination and gene amplification to prevent copy number increases beyond two copies. We are aware that our experimental design does not reflect how evolution may occur in the wild. However, this design allowed us to study evolutionary forces separately that are otherwise difficult to disentangle.

In addition, because evolution experiments need to occur on laboratory time scales, such experiments incorporate design elements that may cause the evolutionary dynamics to differ from that in the wild. Examples include the use of plasmids with copy numbers higher than one, high mutation rates, and intense selection. It is possible that under natural conditions the higher frequency of beneficial mutations in the early evolution of our double-copy populations might lead to a bigger advantage of gene duplication than we observed.

The coGFP protein can be viewed as a simple model of a promiscuous protein, because of its dual color (green and blue) fluorescence. A proposed resolution to Ohno's dilemma is that the 'innovation' usually associated with neofunctionalization actually precedes gene duplication (*Hughes, 1994*; *Piatigorsky and Wistow, 1991*). This means that the ancestral gene, in addition to its main activity, has one or several minor side activities, i.e., it encodes a promiscuous protein. Promiscuity is a widespread property of proteins (*Copley, 2017*; *Copley et al., 2023*; *Khersonsky and Tawfik, 2010*). It might thus play a key role in the critical early stages of post-duplication evolution, where selection for a side activity may help maintain a new gene copy until emerging mutations can further improve this activity (*Näsvall et al., 2012*; *Kondrashov et al., 2002*; *Copley, 2020*). If the two activities show a strong trade-off, improvement of one would entail deterioration of the other. A single gene encoding such a protein suffers from an adaptive conflict between the two activities. Gene duplication may provide an escape from this adaptive conflict, because each duplicate may specialize on one activity (*Des Marais and Rausher, 2008*; *Hughes, 1994*). For coGFP, a trade-off likely exists for fluorescence in these two colors, because improvement of green fluorescence entails a loss of blue fluorescence during evolution (*Figure 3—figure supplement 1*, *Figure 6—figure supplement 2*). We, therefore, expected that during selection for both green and blue fluorescence, one *cogfp* copy in double-copy populations would 'specialize' on green fluorescence whereas the other copy would specialize on blue fluorescence. However, when we analyzed individual population members with two active gene copies we could not find any such specialization (*Figure 6—figure supplement 7*). Moreover, the identified key mutations at positions 147 and 162 have a very low frequency (<1%) in these populations (*Figure 6—figure supplement 1*). Future experiments with different selection strategies might reveal the reasons for this observation and the conditions under which such a specialization can occur.

In summary, we established an experimental system that allowed us to gain time-resolved phenotypic and genotypic data on the evolution of (exactly) two gene copies to directly test Ohno's hypothesis. Our experiments support that gene duplication increases the mutational robustness of gene duplicates. They also demonstrate that this leads to increased genetic diversity. However, we did not observe accelerated phenotypic evolution or specialization for different phenotypes. Further experiments will be necessary to clarify if these phenomena could occur under different conditions.

Importantly, our experimental system could help to answer many further questions about the evolution of duplicated genes. For example, by allowing the expression of duplicates to be controlled independently from each other, it can help answer how differential expression may affect both genotypic and phenotypic evolution after gene duplication. Furthermore, it could be used to study gene

duplication under selection pressures that change in time, favoring one or the other color phenotype at different times. It could also be used to study the co-evolution of regulatory regions and gene coding regions. In sum, our system not only allowed us to validate a long-standing and difficult to test hypothesis about evolution by gene duplication, it also provides a platform for further evolutionary studies.

# Materials and methods

**Key resources table**

| Reagent type (species) or resource | Designation | Source or reference | Identifiers | Additional information |
|---|---|---|---|---|
| Gene (*Cavernularia obesa*) | *Cogfp* | PMID:23468077 | | codon-optimised for *E. coli* |
| Strain, strain background (*E. coli*) | NEB5α | New England Biolabs | | |
| Recombinant DNA reagent | pAND (plasmid) | PMID:24316737 | RRID:Addgene_49377 | |
| Recombinant DNA reagent | pAND-MCS (plasmid) | this paper | RRID:Addgene_223514 | sequence and plasmid available via Addgene |
| Recombinant DNA reagent | pDUP (plasmid) | this paper | RRID:Addgene_223515 | sequence and l plasmid available via Addgene |
| Recombinant DNA reagent | pDUP1 (plasmid) | this paper | RRID:Addgene_223516 | sequence and plasmid available via Addgene |
| Recombinant DNA reagent | pDUP2 (plasmid) | this paper | RRID:Addgene_223517 | sequence and plasmid available via Addgene |
| Commercial assay or kit | Illustra TempliPhi DNA Amplification Kit | GE Healthcare | GE Healthcare #25640010 | |
| Commercial assay or kit | SMRTbell Barcoded Adapter Complete Prep Kit 96 | Pacific Biosciences | PacBio # 100-514-900 | |
| Commercial assay or kit | The SMRTbellTM Damage Repair Kit | Pacific Biosciences | PacBio # 100-486-900 | |
| Commercial assay or kit | AMPure PB Kit | Pacific Biosciences | PacBio # 100-265-900 | |
| Commercial assay or kit | NEBuilder HiFi DNA Assembly Master Mix | New England Biolabs | NEB #E2621 | |
| Commercial assay or kit | EcoRI | New England Biolabs | NEB #R0101S | |
| Commercial assay or kit | NotI | New England Biolabs | NEB #R3189S | |
| Commercial assay or kit | SacI | New England Biolabs | NEB #R3156S | |
| Commercial assay or kit | KpnI | New England Biolabs | NEB #R3142S | |
| Commercial assay or kit | NdeI | New England Biolabs | NEB #R0111S | |
| Commercial assay or kit | NcoI | New England Biolabs | NEB #R0193S | |
| commercial assay or kit | DpnI | New England Biolabs | NEB #R0176S | |
| Chemical compound, drug | MnCl2 | Sigma-Aldrich | Sigma #M3634 | |

## Reagents

We purchased restriction enzymes, NEBuilder HiFi DNA Assembly Master Mix, and T4 DNA ligase from New England Biolabs (NEB). We obtained oligonucleotides from Sigma-Aldrich. We carried out polymerase chain reactions (PCRs) with Hot Start polymerase (MERCK MILLIPORE) and performed colony PCRs with Taq polymerase (NEB). We used the Monarch PCR and DNA Cleanup Kit (NEB) to

purify PCR products, and the QIAprep Spin Miniprep Kit (QIAGEN) for plasmid purification. For visualization of electrophoretic gels, we used SYBR Safe DNA binding dye (Invitrogen Life Technologies, Carlsbad CA, USA).

For cloning and precultures, we used Luria-Bertani (LB) medium (tryptone 10 g/l; yeast extract 5 g/l; NaCl 10 g/l). For plate reader assays and selection experiments, we used M9 Minimal Medium (1×M9 salts, 2 mM MgSO$_4$, 0.1 mM CaCl$_2$,) supplemented with 0.4% glucose and 100 mM HEPES buffer pH 7.5. For the preparation of the electrocompetent cells, we used 2xTY Medium (Tryptone 16 g/l; yeast extract 10 g/l; NaCl 5 g/l). When indicated, we supplemented media with kanamycin (30 µg/ml).

## Bacterial strain

In this study, we used *Escherichia coli* strain NEB5α recA1, endA1, hsdR17, deoR, thi-1, supE44, gyrA96, relA1, Δ(lacZYA-argF), U169 (φ80dlacZΔM15)(NEB).

## Construction of the plasmids expressing coGFP

We constructed a plasmid to express two copies of the *cogfp* gene independently from each other. As a starting point for this construction, we used the pAND plasmid (Addgene #49377; *Stanton et al., 2014*), a kind gift from Christopher Voigt. This plasmid contains a p15A origin of replication, which causes the plasmid to persist at approximately 10 copies/cell. It also encodes the transcriptional repressors LacI and TetR and a kanamycin resistance gene. We removed the NdeI restriction site from TetR, using site-directed mutagenesis with the pair of primers LJM06_f and LJM06_r (*Appendix 1— table 3*). Next, we designed and synthesized (Invitrogen, GeneArt) a multiple cloning site (MCS), a bidirectional terminator BBa_B0014 in the registry of standard biological parts (https://parts.igem. org/Part:BBa_B0014), P$_{tet}$ and P$_{tac}$ operators, as well as restriction sites, to be able to insert the two copies of the *cogfp* gene. To insert the MCS, we first amplified the backbone with the primers LJM07_r and LJM08_f to introduce EcoRI and NotI restriction sites. We then cut the backbone with EcoRI and NotI restriction enzymes. We amplified the MCS with the primers LJM09_f and LJM09_r (*Appendix 1—table 3*), digested it with EcoR and NotI, and ligated it into the backbone to yield pAND-MCS. We then codon-optimised *cogfp* for *E. coli*. We also included the anti-dimerization mutations S129G, C154S, D156G, K204I, and N209Y as previously described (*Ogoh et al., 2013*) and added an N-terminal His-tag. We then let this sequence synthesize (Invitrogen, GeneArt). We cloned the sequence into pAND-MCS at the position 1 (under P$_{tet}$) using the restriction enzymes NdeI and SacI, followed by ligation to yield pDUP. We then amplified *cogfp* with the primers LJM10_f and LJM10_r (*Appendix 1—table 3*) to insert NcoI and KpnI restriction sites. Subsequently, we used NcoI and KpnI to clone an identical copy of *cogfp* at position 2 (under P$_{tac}$) into pDUP to yield pDUP2. To change to a different variant of coGFP (including the inactive version: Q74A, Y75S, G76A), we performed site-directed mutagenesis on a vector containing a single copy of *cogfp* using the primers in *Appendix 1—table 4* and repeated the cloning step just described to insert the second copy. Plasmids used in this study are in *Appendix 1—table 5*.

## Mutagenesis and selection

To generate mutant libraries, we introduced random mutations with Multi Primed Rolling Circle Amplification using the Illustra TempliPhi DNA Amplification Kit (GE Healthcare) with 2 mM of MnCl$_2$ added to a final reaction volume of 10.8 µl. The starting material for the TempliPhi amplifications was plasmid DNA (10 ng) in a 5.5 µl sample mix that contains sample buffer and random hexamers that prime DNA template non-specifically. We heated the sample to 95 °C for 3 min, then cooled and combined it with 5.3 µl of reaction mix that contains salts and deoxynucleotides. Addition of Phi29 polymerase (0.2 µl) initiated the amplification reaction that we incubated for 20 hr at 30 °C. At the end of the incubation, we inactivated the DNA polymerase by incubation at 65 °C for 10 min.

We then digested the TempliPhi reaction products (10.8 µl) using NdeI and NcoI, with addition of DpnI (each 20 U) in NEBuffer CutSmart to a final volume of 20 µl. We performed digestions at 37 °C for 3 hr, and then heated the reaction mix at 65 °C for 10 min to stop the enzyme reaction. We confirmed amplifications and digestions by running 5 µl of the digestion reaction on a gel using agarose gel electrophoresis.

We amplified the plasmid backbone into which the *cogfp* coding regions was to be inserted by PCR, using KOD Hot Start polymerase (95 °C 2 min, 30 cycles of 95 °C 20 s, 60 °C 30 s, 70 °C 2 min

10 s) with the forward oligonucleotide primer LJM01_f and the reverse primer LJM01_r (*Appendix 1—table 3*) from pDUP2 (10 ng) as a template in a reaction volume of 50 µl. After the reaction, we purified the PCR product with Monarch PCR and DNA Cleanup Kit (NEB).

We used Gibson assembly to clone the digested TempliPhi amplicons into the plasmid backbone. This re-insertion could occur in both directions, a gene that was controlled by the Ptet promoter in one generation might come under the control of Ptac in the next generation, and vice versa. We prepared a reaction mixture of 5 µl containing 2 µl of digested amplicons, 7.5 ng of plasmid backbone, and 2.5 µl NEBuilder HiFi DNA Assembly Master Mix (NEB). We incubated the reaction mix for 16 hr at 50 °C. To purify the reaction products by sodium acetate precipitation, we added 2 µl of 3 M sodium acetate, and 50 µl of 100% ethanol. We incubated this mixture at –80 °C for 3 hr and centrifuged it for 15 min at 4 °C. We washed the precipitated DNA samples with 50 µl of 70% ethanol and dried them at 45 °C in a heat block. We resuspended the dried DNA in 5 µl of deionized sterile (MilliQ) water and used this suspension for electroporation.

We realized that bi-directional insertion by Gibson assembly led to the conversion of the NdeI site downstream of $P_{tac}$ to a NcoI site (as downstream of the $P_{tet}$) (*Figure 1—figure supplement 2*). We confirmed that this change did not affect expression levels. We, therefore, used a backbone with two NcoI sites instead of a NdeI and a NcoI from generation 2 onwards.

## Preparation of electrocompetent *E. coli* cells

To prepare electrocompetent *E. coli* NEB5$\alpha$ cells, we picked a single colony from a LB-Kan agar plate of freshly streaked cells and inoculated them into 5 ml 2xTY medium. We incubated this starter culture at 37 °C for 3 hr, with shaking at 180 rpm. We then transferred 4 ml of the culture to 2 l flasks containing 250 ml 2xTY medium and incubated the cells at 37 °C until the culture reached an $OD_{600}$ of 0.6. We performed all subsequent steps on ice, and with chilled solutions. We centrifuged the cell suspension at 5000 g for 10 min at 4 °C, removed the supernatant, and resuspended the cell pellet in 50 ml Milli-Q water. We repeated this procedure three times while reducing the resuspension volume at each washing step. This resulted in approximately 1 ml of concentrated cell suspension that we divided into 80 µl aliquots to be used directly for transformation of the mutant libraries.

## Transformation of *E. coli* with the mutant libraries

2 µl of the precipitated and diluted (1:3) Gibson Assembly product was transformed into electro-competent *E. coli* NEB5α cells using a standard electroporation protocol (Eppendorf Electroporator, 2.5 KV, 200 Ohms) with 2 mm cuvettes (Axon Lab). After electroporation 1 ml of SOC medium was added to the transformed cells, which were then incubated at 37 °C for 1 hr at 180 rpm. Next, 50 µl of transformants were plated out on LB-Kan agar plates to determine the library size. The rest of the transformed culture (950 µl) was transferred into 9 ml of LB-Kan, grown overnight, and used to make glycerol (15% w/v) stocks in aliquots of 700 µl and stored at –80 °C.

## Microplate reader measurements

To quantify the level of the fluorescence of the libraries, we thawed their glycerol stocks on ice; collected cells by centrifugation (2500 rpm, 4 min) and washed them with 500 µl of M9-Kan medium supplemented with 0.02 µM aTc and 100 M IPTG and diluted to $OD_{600}$ 0.25. We then transferred 120 µl of the libraries to 96-well plates with lids and incubated it at 37 °C in a BioTek plate reader (Synergy H1 microplate reader) for 15 hr. During the incubation, we measured absorbance at 600 nm, green fluorescence (excitation: 388 nm, emission: 507 nm), and blue fluorescence (excitation: 388 nm, emission: 456 nm), every 10 min while shaking the plates continuously between the readings (double orbital, 2 mm). After 15 hr of incubation, we performed spectral scanning of coGFP variants in the cell populations by exciting them at 388 nm and measuring the emission spectra (420 nm to 600 nm).

## Selection of mutant libraries using FACS

We cultured libraries for 15 hr in the microplate reader as described above ('Microplate reader measurements'), collected cells by centrifugation (2500 rpm, 4 min) in Eppendorf tubes, and resuspended them in 1 ml PBS. Finally, cells were diluted fivefold with PBS before the sort. We sorted libraries using an Aria III cell sorter (BD Biosciences), with a 70 nm nozzle and a flow rate set to 2 µl/s. We used the AmCyan (407 nm excitation, 530/30 nm emission) and DAPI (407 nm excitation, 450/40 nm emission)

channels to measure green and blue fluorescence, respectively. To define the selection regimes (gates) for cell sorting, we used fluorescence scatter plots with the fluorescence color of interest (*Figure 1—figure supplement 4*). We used a population of cells that carry the plasmid pDUP1 but do not express coGFP (uninduced), as a negative control that allowed us to estimate the autofluorescence of the bacterial cells.

For each gate, we sorted 20'000 cells into a 6 ml tube containing 500 µl of LB medium. After sorting, we added 1 ml more of LB medium to the tubes, and incubated the cells at 37 °C without antibiotic and without shaking for 30 min. We continued the incubation for an additional 2 hr but with shaking at 220 rpm.

Next, we transferred the sorted cells into 10 ml LB-Kan and incubated them overnight at 37 °C in 50 ml tubes with shaking at 220 rpm. On the next day, we prepared 700 µl aliquots of the overnight cultures and stored them as glycerol stocks (15% w/v) at –80 °C. From the rest of the overnight cell cultures, we extracted plasmids that we used as a templates for the next round of mutagenesis.

## Analysis of the evolved populations using flow cytometry

After each round of sorting, we analyzed the phenotypes of the evolved populations and quantified their fluorescence level using flow cytometry. To this end, the glycerol stocks of the sorted populations were thawed on ice. The libraries were grown for 15 hr in a microplate reader (as described in 'Microplate reader measurements'). Afterwards, cultures were collected by centrifugation (2500 rpm, 4 min), washed with 1 ml PBS buffer, and diluted 20 times with PBS before analysis. Phenotypes of all mutant libraries and evolved populations were re-analyzed using Cytoflex BC (The Beckman Coulter Life Sciences). We used PB450 channel (405 nm excitation, 450/55 nm emission) and KO525 channel (405 nm excitation, 525/40 nm emission) to measure blue and green fluorescence, respectively.

## Flow cytometry data analysis

We used FlowJo (FlowJo, LLC) for flow cytometry data analysis and displayed the data on biexponential scales. We first used forward scatter height (FSC-H) versus side scatter height (SSC-H) density plots to gate cell populations. Next, AmCyan-Height (AmCyan-H) and DAPI-Height (DAPI-H) data were exported to be analyzed in R. For each evolving population, we used AmCyan-Height (AmCyan-H) versus DAPI-Height (DAPI-H) density plots to further gate fluorescent cells that displayed a higher fluorescence than the non-induced control carrying the ancestral plasmid pDUP2 as indicated in *Figure 2—figure supplement 1*. For the mutational robustness analysis (*Figure 2*), we analyzed the libraries after mutagenesis before sorting. We indicate the percentage of all cells falling outside the gate of the negative control. For the analysis of the fluorescence (*Figure 3*), we report the median fluorescence intensity of the populations recovered after sorting.

## Library preparation for PacBio single molecule real-time sequencing (SMRT, Pacific Biosciences)

We thawed glycerol stocks of libraries on ice. For each library, we added 350 µl of the glycerol stock to 5 ml LB-Kan and incubated it at 37 °C overnight. We then performed plasmid extractions from the overnight cell cultures using the QIAprep Spin Miniprep Kit (QIAGEN) for plasmid purification. We digested 3 µg of plasmid DNA from each of the library samples using the restriction enzymes EcoRI and NotI (20 U each), and CutSmart NEBuffer, in a total reaction volume of 50 µl. The digestions were performed at 37 °C for 2 hr. Afterwards, we heated the digestion reaction at 65 °C for 20 min to deactivate the enzymes. We confirmed successful digestion by analyzing 5 µl of the digestion mix using agarose gel electrophoresis, and purified the rest of the digestion product with Monarch Nucleic Acid Purification Kit (NEB). We quantified the amount of purified DNA using the Qubit dsDNA HS Assay (Thermo Fisher) and Qubit fluorometer 3.0 (Thermo Fisher).

Next, we prepared SMRTbell libraries using the SMRTbell Barcoded Adapter Complete Prep Kit 96 (PacBio # 100-514-900) following the manufacturer's recommendations (PacBio protocol number 100-538-700-03). Briefly, we used 80 ng of each library in the end-repair/ligation reaction using barcoded adaptors from the SMRTbell barcoded adapter complete prep kit –96 (PacBio # 100-514-900). We then pooled all the barcoded libraries to be multiplexed and purified the pooled library using the AMPure PB Kit (PacBio # 100-265-900). We used a concentration of 0.6 X of AMPure Beads and eluted the DNA in 37 µl of elution buffer. After pooling, we performed a DNA damage repair

reaction followed by exonuclease treatment (ExoIII and ExoVII) using the SMRTbell DNA Damage Repair Kit (PacBio # 100-486-900) according manufacturer's instructions. Finally, we purified the DNA twice with AMPure Beads as before and eluted the DNA in 15 µl of the provided elution buffer.

We performed a size selection of the pooled barcoded libraries to isolate DNA molecules between 1.4 and 2.4 kb in length, using a BluePippin system (Sage Science, Inc Beverly, MA, USA). The fragment of interest was checked on a Fragment Analyzer (Agilent Technologies, Santa Clara, CA, USA). We performed sequencing in two runs. For the first batch of samples, we used v3.0/v3.0 chemistry and diffusion loading on a PacBio Sequel I instrument (Pacific Biosciences, Menlo Park, CA, USA) at 600 min movie length, pre-extension time of 30 min using one SMRT cell 1M v3. For the second batch, we sequenced on 1 SMRT cell 8 M with v2.1/v2.0 chemistry on a PacBio Sequel II instrument (Pacific Biosciences, Menlo Park, CA, USA) at a movie length of 15 hr. Both runs were performed by the Lausanne Genomic Technologies Facility.

## Engineering specific mutants

We used 10 ng of a variant plasmid pDUP, in which the *cofgp* gene is regulated by P$_{tac}$ (i.e. it is IPTG inducible), as a template for site-directed mutagenesis of *cogfp*. We carried out PCRs with KOD Hot Start polymerase (95 °C 2 min, 16 cycles of 95 °C 20 s, 60 °C 30 s, 70 °C 2 min) using the primers listed in *Appendix 1—table 4*. We then treated the amplification product with DpnI (10 U) and purified it with the Monarch Nucleic Acid Purification Kit. We used 4 µl of the purified product for the transformation of the electrocompetent NEB5alpha cells using a standard electroporation protocol. After electroporation, we added 1 ml of SOC medium to the transformed cells and incubated them at 37 °C for 1 hr at 180 rpm. We then plated 200 µl of cells on LB-Kan agar plates. We engineered multi-mutation variants by introducing the mutations serially using site-directed mutagenesis. We Sanger sequenced plasmids and we stored them as glycerol (15% w/v) stocks in aliquots of 700 µl at –80 °C.

## Protein purification of specific mutants

We purified coGFP protein variants using His-tag affinity purification. From the glycerol stocks of each mutant, we inoculated 2 ml LB-Kan, and cultured them overnight at 37 °C. The following day we inoculated 50 µl of the overnight culture into 10 ml of fresh LB-Kan medium and 100 µM IPTG to induce protein expression. We grew the cultures at 30 °C for 16 hr before pelleting cells at 10,000 g for 8 min. We immediately froze the cell pellets and stored them at –20 °C.

We chemically lysed cell pellets in 200 µl of freshly prepared lysis buffer containing Cell lytic Buffer-B (Sigma #B7435), 100 mM Tris/HCl pH 7.5, 150 mM NaCl, supplemented with 0.2 mg/ml lysozyme (Sigma #10837059001), 50 units/ml benzonase (Sigma #E1014) and 1 x Protease Inhibitor Cocktail (cOmplete EDTA-free, Roche). We incubated re-suspended pellets at room temperature for 1 hr with gentle shaking (300–400 rpm). We centrifuged crude cell lysates at 10,000 g for 10 min at 4 °C and transferred the supernatant into fresh 1.5 ml Eppendorf tubes. To each tube we added 200 µl of pre-equilibrated 50% His Mag Sepharose beads slurry (Cytiva #28967390; pre-equilibrated three times in 1 ml equilibration buffer (20 mM Tris pH 7.5, 500 mM NaCl, 20 mM imidazole) for 5 min each) and incubated the lysate-Magbead mix for 1 h on a spinning wheel at 4 °C. Subsequently, we used a magnetic rack to remove the supernatant and wash the beads three times for 5 min each with 500 µL equilibration buffer. Finally, we eluted the His-tagged protein with 100 µl elution buffer (20 mM Tris pH 7.5, 500 mM NaCl, 500 mM imidazole) for 15 min. To calculate the protein concentration, we measured absorbance at 280 nm, using an extinction coefficient of 19730 M$^{-1}$cm$^{-1}$, calculated based on the coGFP sequence, using an online calculator (available here). We confirmed the purity of proteins by SDS-PAGE showing only a single band corresponding to coGFP for all variants (>95% purity).

## Fluorescence spectra of purified coGFP variants

To measure fluorescence spectra of engineered coGFP variants, we used freshly purified protein (0.05 mg/ml in elution buffer). For each engineered coGFP variants, we performed measurements in triplicates with 120 µl of protein solution per well in 96-well plates, with an excitation wavelength of 388 nm and an emission bandpass filter of 430–550 nm (Biotek, Synergy H1).

## Western blot of coGFP variants

To determine changes in expression level and solubility of coGFP variants we performed SDS-PAGE (Sodium dodecyl-sulphate polyacrylamide gel electrophoresis) followed by immunostaining for His-tagged coGFP of clarified cell lysate. In preparation for gel electrophoresis, we inoculated cell cultures from the glycerol stocks into 2 ml LB-Kan medium and incubated overnight at 37 °C. To induce CoGFP expression, we used 40 µl of the overnight culture, inoculated 2 ml of the fresh LB-Kan medium containing 100 µM IPTG, and incubated at 30 °C for 16 hr. We centrifuged cell cultures at 10,000 g for 8 min to collect the cell pellet. We lysed cell cultures using 100 µl lysis buffer and clarified them by spinning at 10,000 g for 10 min at 4 °C. We transferred 30 µl of the supernatant into 1.5 ml Eppendorf tube and mixed with 4 x SDS-PAGE sample buffer containing 0.1 M fresh Dithiothreitol (DTT) and heated for 10 min at 95 °C. We used a precast gel (Biorad, 10% Mini-Protean, 10 wells) to load 15 µl of sample per well and a Biorad PAGE system for 5 min at 50 V, followed by 60 min at 100 V in Laemmli running buffer to separate them. Subsequently, we plotted gels onto a nitrocellulose membrane for 1 h at 100 V at 4 °C in Western Blot buffer (25 mM Tris, pH 8.3, 192 mM glycine, 1% SDS, 20% ethanol). After the transfer, the membrane was washed with milliQ water, and blocked for at least 1 hr with 5% Bovine Serum Albumin (BSA) in PBS-T buffer (PBS pH 7.5 with 0.1% Tween-20).

Next, we performed immunostaining to detect His-coGFP on the nitrocellulose membrane. We used Penta-His mouse monoclonal antibody (Qiagen, Qiagen #34660) at a 1:2000 dilution in 5% BSA in PBS-T buffer for 1 hr, followed by three washes with the PBS-T buffer, 15 min each. We added a secondary horseradish peroxidase (HRP) conjugate antibody (goat anti-mouse, Sigma #12–349) at a 1:5000 dilution in 5% BSA in PBS-T, followed by three washes with PBS-T buffer for 10 min each. After washing, we added an enhanced chemiluminescence (ECL, Biorad) solution. We used a Fusion Fx7 imaging system (Peqlab) for the visualization of the membrane.

## Size-exclusion chromatography (SEC) analysis

We performed size-exclusion chromatography of His-tag purified coGFP variants on an Äkta Pure chromatography system (GE Healthcare), with a Superdex 75 10/300 Increase column (GE Healthcare) at a flow rate of 0.5 mg/ml. The running buffer consisted of 20 mM Tris/HCl pH 7.5 with 100 mM NaCl. We used the fractions of folded protein, to measure protein fluorescence at 388 nm excitation, 457 nm (blue), and 507 nm (green) emission, using a plate reader (Biotek, Synergy H1). We normalized protein concentrations to 0.01 mg/ml and measured protein fluorescence in triplicates in a 200 µl volume.

## Modeling of coGFP mutations

Computational models of the coGFP wild-type protein were based on the coGFP uniprot sequence (https://www.uniprot.org/uniprot/T2HNK0) with the anti-dimerization mutations S129G, C154S, D156G, K204I, and N209Y (*Ogoh et al., 2013*) and additionally S147G. The models were built using the following templates:

- Green Fluorescent Protein from *Renilla reniformis*, (Protein Data Base [PDB] id: 2RH7), with 51% of sequence identity to coGFP (*Loening et al., 2007*).
- GFP from Philippine soft coral (PDB id: 4JC2), with 46% of sequence identity to coGFP

100 models of each variant (single, double, and triple mutants) were calculated with modeler 9.18 v. The chromophore structure was modeled as a rigid body.

Protein stability changes between mutated and ancestral proteins were calculated for models using the FoldX 4.0 software (*Schymkowitz et al., 2005*). Each mutation, as well as combinations of 2 and 3 mutations, were reversed to the ancestral sequence for each structure, and the predicted changes in structural stability were assessed. In the case of mutations G147S and V162D, only the structures with rotamers that allow formation of hydrogen bonds between G147S and chromophore residues, as well as between G147S and V162D were taken into account in the calculations (30–40 structures per mutant).

Structural stability (referred to as ΔG) is the difference in free energy between a protein in the folded and the unfolded state. Values of ΔG were calculated and compared for the ancestral protein and the mutated proteins. Therefore, we refer to the change in protein stability due to mutations as a change in the ΔG - the ΔΔG. A positive value of ΔΔG indicates a destabilizing effect of a mutation

or set of mutations, whereas a negative value indicates a stabilizing effect of mutation. Additionally, the sum of impacts of single mutations was estimated to assess potential synergistic effect of the mutations. Figures of modeled structures were prepared with UCSF Chimera software (*Pettersen et al., 2004*).

## Processing of the sequencing data

PacBio SMRT sequencing reads the same circularized DNA molecule several times, i.e., in several 'passes' through the DNA. The proprietary software smrtlink (version 9) identifies a pass by the presence of an adapter sequence. Since our target DNA sequence is palindromic (double-copy of *cogfp* in opposite orientation), smrtlink cannot differentiate a single pass through the palindromic DNA from two passes of a single copy of *cogfp*. To identify the passes correctly, we used a recall adapter (https://github.com/PacificBiosciences/recalladapters, *Pacific Biosciences, 2022*) to manually splice adapter sequences in silico. The resulting data files could then be correctly processed using smrtlink. This processing involves two steps. The first is consensus calling, which generates a consensus sequence from multiple sequencing passes. This step reduces sequencing errors. The second step is demultiplexing, where different reads are assigned to different experimental samples based on barcode sequences. This step also converts the data from bam format to fasta format.

We aligned the resultant fasta files to the reference sequence using the program minimap2 (SMRT tools, PacBio). For subsequent analyses, we only used the reads that aligned completely with the reference sequence (full query coverage). We note that unaligned reads may result from contamination by *E. coli* genomic DNA. Furthermore, we reverse-complemented the reads that aligned with the reference sequence in the plus-minus orientation (i.e. reads that are the reverse complement of the reference sequence before alignment), so that all reads have the same orientation relative to the reference sequence. Such reads arise because SMRT sequencing reads DNA from a circularized template in both directions.

We used the reads thus filtered for a second round of alignment. Specifically, we performed a global alignment (*Needleman and Wunsch, 1970*) to map the reads to the reference sequence, using the program needle (*Rice et al., 2000*) to produce an alignment file in *srspair* format. A global alignment is necessary for an end-to-end alignment to the reference sequence, which cannot be achieved through a local alignment algorithm such as minimap2.

An inactive copy should contain the three specific mutations that we had engineered in place of the amino acids that form the fluorophore (Q74A, Y75S, G76A). However, we reasoned that absence of purifying selection on the inactive copy can cause the engineered mutations to be replaced by other mutations. Therefore, we designated a *cogfp* copy as inactive if it met any one of the following three criteria: First, the 75[th] amino acid in the protein sequence is not tryptophan, phenylalanine, or tyrosine. Second, the 76[th] amino acid is not glycine. Third, a premature stop codon exists anywhere between codon positions 1 and 220.

Next, we computed the frequencies of single amino-acid mutations and of multi-mutant genotypes in each copy of the *cogfp* gene using custom scripts written in the awk programming language. We did so for *cogfp* variants isolated after each round of evolution, for each replicate experiment, and for each of our six selection strategies. Our subsequent analyses focused on non-synonymous mutations, because they are more likely to alter the spectral properties of coGFP.

## Analysis of sequence divergence

We used three metrics to understand the sequence evolution of our coGFP variants. The first is the number of amino-acid mutations relative to ancestral coGFP, which indicates how much the variants have diverged from the ancestor. The second metric is the average pairwise distance, which is the average number of amino acid differences between any two coGFP variant sequences (*Iyengar and Wagner, 2022a*; *Iyengar and Wagner, 2022b*). Specifically, this metric is defined as:

$$\frac{1}{n\left(n-1\right)} \sum_{i}^{n} \sum_{j \neq i}^{n} \delta_{ij}$$

where $\delta_{ij}$ denotes the number of amino acid differences between sequence i and sequence j, and n is the total number of sequences sampled from an evolving population. This metric reflects the diversity of protein sequences within the population.

The third metric is the number of non-synonymous mutations relative to that of synonymous mutations, that is the dN/dS ratio. This ratio is defined as:

$$\frac{number\ of\ nonsynonymous\ mutations}{number\ of\ nonsynonymous\ sites} \div \frac{number\ of\ synonymous\ mutations}{number\ of\ synonymous\ sites}$$

## Differential enrichment of mutations

To understand if specific mutations are associated with specific phenotypes, we calculated their allele frequencies in different evolving populations. For example, mutations with significantly higher frequency in *blue-only* populations relative to *green-only* populations may be responsible for a color shift towards blue fluorescence. In a first step, we performed this analysis for single-copy libraries selected only for *blue* or *green* fluorescence. The reason is that the fluorescence of populations with two intact gene copies can result from the fluorescence of two different *cogfp* variants expressed in the same cell, which makes it difficult to establish which mutations are responsible for the phenotype.

To focus on mutations with the strongest influence on fluorescence, we restricted this analysis to mutations that reached the frequency of at least 5% after five rounds of evolution, in at least one replicate population selected for *blue-only* or *green-only*. We then used generalized linear models (GLM) to identify mutations that are significantly more frequent at the end of evolution in populations selected only for blue fluorescence than in populations selected only for green fluorescence, and vice versa. Specifically, we fitted a GLM with a binomial model (logit link function), using mutation counts as a response variable, and selection strategy (*blue-only* and *green-only*) as the predictor variable. Next, we performed an analysis of deviance with a likelihood ratio test using the function ANOVA (R stats package v3.4.4). This function analyses if the fitted GLM is significantly better than a null model where mutation frequencies are not affected by the selection strategy. We adjusted the resulting p-values using a Bonferroni correction for multiple testing. We identified mutations as differentially enriched if their Bonferroni-corrected p-values are lower than 0.05.

## Acknowledgements

We thank Julien Marquis and the team from the Lausanne Genomics Technology Facility for carrying out the SMRT sequencing. We thank M Robinson-Rechavi and F Guillaume for helpful scientific discussions.

## Additional information

### Funding

| Funder | Grant reference number | Author |
|---|---|---|
| Schweizerischer Nationalfonds zur Förderung der Wissenschaftlichen Forschung | 310030_200532 | Yolanda Schaerli |
| Schweizerischer Nationalfonds zur Förderung der Wissenschaftlichen Forschung | 31003A_175608 | Yolanda Schaerli |
| Schweizerischer Nationalfonds zur Förderung der Wissenschaftlichen Forschung | 31003A_172887 | Andreas Wagner |

| Funder | Grant reference number | Author |
|---|---|---|
| European Research Council | 10.3030/739874 | Andreas Wagner |
| Fondation Herbette | | Yolanda Schaerli |
| University of Zurich | URPP Evolution in Action | Yolanda Schaerli |

The funders had no role in study design, data collection and interpretation, or the decision to submit the work for publication.

## Author contributions
Ljiljana Mihajlovic, Conceptualization, Formal analysis, Investigation, Visualization, Methodology, Writing – original draft, Writing – review and editing; Bharat Ravi Iyengar, Data curation, Software, Formal analysis, Visualization, Methodology, Writing – review and editing; Florian Baier, Formal analysis, Investigation, Writing – review and editing; Içvara Barbier, Formal analysis, Writing – review and editing; Justyna Iwaszkiewicz, Software, Formal analysis, Writing – review and editing; Vincent Zoete, Resources; Andreas Wagner, Conceptualization, Supervision, Funding acquisition, Writing – original draft, Writing – review and editing; Yolanda Schaerli, Conceptualization, Supervision, Funding acquisition, Writing – original draft, Project administration, Writing – review and editing

## Author ORCIDs
Ljiljana Mihajlovic ⬤ http://orcid.org/0009-0007-5921-9831
Andreas Wagner ⬤ https://orcid.org/0000-0003-4299-3840
Yolanda Schaerli ⬤ https://orcid.org/0000-0002-9083-7343

Reviewer #1 (Public review): https://doi.org/10.7554/eLife.97216.3.sa1
Reviewer #2 (Public review): https://doi.org/10.7554/eLife.97216.3.sa2
Author response https://doi.org/10.7554/eLife.97216.3.sa3

# Additional files

## Supplementary files
MDAR checklist

Source data 1. Summary of single-molecule real-time (SMRT) sequencing results. Number of reads sequenced by SMRT sequencing and mean number of amino-acid changes per *cogfp* gene. X, Y, Z: replicate populations. 1–5: generations of evolution.

## Data availability
SMRT sequencing data are available at NCBI Sequence Read Archive (SRA) under the BioProject ID PRJNA995516. Codes used for sequencing data analysis are available on GitHub (copy archived at *SchaerliLab, 2025*). Plasmids and their annotated sequences are available at Addgene under the following numbers: 223514–223517. Source data files contain the numerical data used to generate Figures 2-6.

The following dataset was generated:

| Author(s) | Year | Dataset title | Dataset URL | Database and Identifier |
|---|---|---|---|---|
| Mihajlovic L, Iyengar BR, Wagner A, Schaerli Y | 2023 | Directed evolution of a duplicated fluorescent protein | https://www.ncbi.nlm.nih.gov/bioproject/?term=PRJNA995516 | NCBI BioProject, PRJNA995516 |

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

# Appendix 1

**Appendix 1—table 1.** Values of structural stability difference DDG between homology models of coGFP structures and the mutated structures, calculated with Buildmodel FoldX function.
The results pointing to destabilizing effects are colored orange and stabilizing or synergistic are colored green.

| Mutation | DDG by FoldX (kcal/mol) | å of individual single mutant DDG (kcal/mol) | Difference |
|---|---|---|---|
| L98M | −0.25 | −0.25 | 0 |
| G147S | −0.57 | −0.57 | 0 |
| V162D | +3.46 | +3.46 | 0 |
| G147S_V162D | +3.33 | +2.86 | +0.47 |
| L98M_G147S_V162D | +1.95 | +2.64 | −0.69 |
| L98M_V162D | +3.28 | +3.21 | 0.08 |
| L98M_G147S | −1.05 | −0.82 | −0.23 |

Supplementary notes to *Appendix 1—table 1*. The G147S mutation alone is stabilizing the structure because of the hydrogen bond that it is forming with the chromophore Y75 residue (*Appendix 1—table 1*). The mutation of L98M is also changing favorably the structural stability due to longer Met side chain, that is able to form stabilizing hydrophobic interactions with V64 residue of the central helix of coGFP (*Appendix 1—table 1*, *Figure 6—figure supplement 6B*). The L98 residue's side chain is too short to contact the hydrophobic residues of the central helix of coGFP (*Figure 6—figure supplement 6A*).

The V162D is destabilizing the structures, probably due to the presence of residue E194 in a close proximity of it and a potential electrostatic repulsion between these two equally charged side chains (*Appendix 1—table 1*). It is however possible, that pronounced destabilizing character of V162D mutation might be to some extent an artefact of the method, as the coGFP protein is negatively charged and every mutation increasing the negative charge, like V162D, is estimated as very unfavorable to the stability, whereas in reality, its local effects might not be that substantial. As a result, the potentially slightly destabilizing mutation of V162D is overestimated.

These results are in agreement with enrichments analysis (*Figure 6—figure supplement 1*) as the V162D mutation is observed frequently only under green fluorescence selection, otherwise this mutation is not frequent. On the other hand, the L98M mutation is being observed frequently in all populations under all selection regimes, thanks to its structurally stabilizing role.

When paired with V162D, the otherwise stabilizing G147S mutation has, surprisingly, destabilizing effect, despite the possibility of hydrogen bond formation with the aspartate residue 162. It is in agreement with the fraction of folded protein measured for V162D_G147S that was lowest from all of the measured proteins (*Figure 6—figure supplement 5*). However, in case of the triple mutant, V162D, G147S, L98M, the protein has a higher structural stability than the sum of the single mutations' ΔΔG values, pointing to synergistic effect of these mutations (*Appendix 1—table 1*) that is in agreement with the triple mutant's improved folding ability (*Figure 6—figure supplement 5*).

**Appendix 1—table 2.** Detailed statistics for data reported in *Figure 6*.
The test is based on a generalized linear model (binomial model, mutation counts vs library type, likelihood ratio test). Beta means enrichment, p-values are bonferroni corrected. Negative beta means enriched in single-copy populations compared to double-copy populations. Positive beta means enriched in double-copy populations compared to single-copy populations.

| Mutation | Generation | Median frequency in single-copy populations | Median frequency in double-copy populations | p-value | beta |
|---|---|---|---|---|---|
| G147S | 1 | 0.38 | 0.52 | 1.00E+00 | 0.38 |
| G147S | 2 | 0.51 | 2.01 | 2.21E-112 | 1.51 |

*Appendix 1—table 2 Continued on next page*

*Appendix 1—table 2 Continued*

| Mutation | Generation | Median frequency in single-copy populations | Median frequency in double-copy populations | p-value | beta |
|---|---|---|---|---|---|
| G147S | 3 | 0.07 | 5.55 | 0.00E+00 | 3.95 |
| G147S | 4 | 7.89 | 1.26 | 1.00E+00 | −0.02 |
| G147S | 5 | 5.58 | 12.98 | 0.00E+00 | −1.81 |
| V162D | 1 | 0.11 | 0.30 | 1.49E-02 | 1.31 |
| V162D | 2 | 1.98 | 5.85 | 4.99E-244 | 1.06 |
| V162D | 3 | 92.13 | 89.02 | 1.26E-01 | 0.06 |
| V162D | 4 | 99.41 | 98.74 | 6.21E-12 | −1.20 |
| V162D | 5 | 98.99 | 99.23 | 4.25E-03 | −0.41 |
| L98M | 1 | 0.68 | 0.93 | 5.23E-01 | 0.53 |
| L98M | 2 | 4.90 | 4.50 | 1.00E+00 | −0.02 |
| L98M | 3 | 4.27 | 8.29 | 2.27E-122 | 0.58 |
| L98M | 4 | 5.44 | 20.00 | 4.28E-210 | 2.11 |
| L98M | 5 | 10.00 | 11.13 | 7.96E-138 | 0.59 |
| L98M+G147 S | 1 | 0.00 | 0.00 | 1.00E+00 | 1.04 |
| L98M+G147 S | 2 | 0.01 | 0.00 | 4.79E-02 | −1.65 |
| L98M+G147 S | 3 | 0.00 | 0.00 | 9.42E-141 | 4.11 |
| L98M+G147 S | 4 | 0.00 | 0.69 | 5.29E-08 | 2.51 |
| L98M+G147 S | 5 | 0.64 | 4.24 | 0.00E+00 | −2.28 |
| L98M+V162D | 1 | 0.00 | 0.00 | 1.00E+00 | 1.04 |
| L98M+V162D | 2 | 0.00 | 0.00 | 1.00E+00 | −0.67 |
| L98M+V162D | 3 | 0.06 | 4.02 | 0.00E+00 | 3.22 |
| L98M+V162D | 4 | 5.08 | 19.43 | 2.91E-186 | 2.09 |
| L98M+V162D | 5 | 9.67 | 10.95 | 1.11E-114 | 0.54 |
| G147S+V162D | 1 | 0.00 | 0.00 | 1.00E+00 | 1.32 |
| G147S+V162D | 2 | 0.00 | 0.00 | 1.00E+00 | 0.25 |
| G147S+V162D | 3 | 0.00 | 0.00 | 1.00E+00 | −0.34 |
| G147S+V162D | 4 | 7.89 | 0.69 | 1.00E+00 | −0.16 |
| G147S+V162D | 5 | 5.58 | 12.91 | 0.00E+00 | −1.82 |
| L98M+G147S+V162D | 1 | 0.00 | 0.00 | 1.00E+00 | 1.04 |
| L98M+G147S+V162D | 2 | 0.00 | 0.00 | 1.00E+00 | 0.25 |
| L98M+G147S+V162D | 3 | 0.00 | 0.00 | 1.00E+00 | 0.17 |
| L98M+G147S+V162D | 4 | 0.00 | 0.57 | 8.99E-08 | 2.49 |
| L98M+G147S+V162D | 5 | 0.64 | 4.24 | 0.00E+00 | −2.28 |

**Appendix 1—table 3.** Primers used in this study.

| Oligonucleotide | Oligonucleotide sequence (5 → 3) |
|---|---|
| Seq_0_f | GAGTTGTAAAACGACGGCCAG |

*Appendix 1—table 3 Continued on next page*

*Appendix 1—table 3 Continued*

| Oligonucleotide | Oligonucleotide sequence (5 → 3) |
| --- | --- |
| Seq_2_r | GAAAGCTGGTCCAAGCGATTG |
| Seq_3_f | CTCATTCGCTAATCGCCAC |
| pBAD_f | GCCGTCACTGCGTCTTTTAC |
| LJM01_f | GTG ATG ATG GTG ATG ATG GCC CAT ATG TAT ATC TCC |
| LJM01_r | GTG ATG ATG GTG ATG ATG GCC CAT GGT ATA TCT CCT |
| LJM02 | GAT ATA CAT ATG GGC CAT CAT CAC CAT CAT CAC AGC ATT CCG GAA AAT |
| LJM03_r | GTTACCAAACTGGAACCGGCGAGCGAAAGCATGTATGTTAG |
| LJM03_f | CTAACATACATGCTTTCGCTCGCCGGTTCCAGTTTGGTAAC |
| LJM04_f | GTTACCAAACTGGAACCGGGCAGCGAAAGCATGTATGTTAG |
| LJM04_r | CTAACATACATGCTTTCGCTGCCCGGTTCCAGTTTGGTAAC |
| LJM05 | GGC CAT CAT CAC CAT CAT CAC |
| LJM06_f | CTTATTCGGCCTTGAATTGATTATATGCGGATTAGAAAAACAACT |
| LJM06_r | AGTTGTTTTTCTAATCCGCATATAATCAATTCAAGGCCGAATAAG |
| LJM07_r | CAACTCGAATTCTTCCACCGTACGTCGAGCGGGAG |
| LJM08_f | GATATAGCGGCCGCAATGGCGGCGCGCCATCGAATG |
| LJM09_f | GTCATGGAATTCGAGTTGTAAAACGACG |
| LJM09_r | GATTATGCGGCCGCGCCGTCACTGCGTCTTTTAC |
| LJM10_f | GCTAGC CCATGG GCCATCATCATCACCATCATAG |
| LJM10_r | CTCTAC GGTACC TTATTACGGTTTGGCAATTGCGGTTTC |

**Appendix 1—table 4.** List of site-directed mutagenesis primers.

| Mutation | Forward primer | Reverse primer |
| --- | --- | --- |
| Q74A, Y75S, G76A | GATATTCTGAGCGTTGCATTT GCC AGC GCG AATCGTACCTATACCAGCTATC | GATAGCTGGTATAGGTACGATT CGC GCT GGC AAATGCAACGCTCAGAATATC |
| V169D | GGTGAAGATGTTCTGAGCTATAAAACCCAGAG CACCCATT | CAGAACATCTTCACCAACCAGGGTGCCATCAC TAACATAC |
| P142L | GATGGTCTGGTTATGAAAAAAGAAGTTACCAA ACTGGAAC | CATAACCAGACCATCTTCCGGGAAACCTTCAC CGTTATAT |
| Y173F | CTGAGCTTTAAAACCCAGAGCACCCATTATAC CTGTCACA | GGTTTTAAAGCTCAGAACAACTTCACCAACCA GGGTGCCA |
| S9R | CATCACCGCATTCCGGAAAATAGCGGTCTGAC CGAAGAAA | CGGAATGCGGAATGCTGTGATGATGGTGATGA TGGCCCAT |
| S9I | CATCACATCATTCCGGAAAATAGCGGTCTGAC CGAAGAAA | CGGAATGATGAATGCTGTGATGATGGTGATGA TGGCCCAT |
| S9C | CATCACTGCATTCCGGAAAATAGCGGTCTGAC CGAAGAAA | CGGAATGCAGAATGCTGTGATGATGGTGATGA TGGCCCAT |
| L105M | CGTACCATGAGCTTTGAAGATGGTGCCATTGT TAAAGTGG | AAAGCTCATGGTACGTTCAAAGGTAAAACCTT CCGGAAAG |
| G154A | GAACCGGCCAGCGAAAGCATGTATGTTAGTGA TGGCACCC | TTCGCTGGCCGGTTCCAGTTTGGTAACTTCTT TTTTCATA |
| G154C | GAACCGTGCAGCGAAAGCATGTATGTTAGTGA TGGCACCC | TTCGCTGCACGGTTCCAGTTTGGTAACTTCTT TTTTCATA |
| G154S | GAACCGAGCAGCGAAAGCATGTATGTTAGTGA TGGCACCC | TTCGCTGCTCGGTTCCAGTTTGGTAACTTCTT TTTTCATA |
| G154D | GAACCGGACAGCGAAAGCATGTATGTTAGTGA TGGCACCC | TTCGCTGTCCGGTTCCAGTTTGGTAACTTCTT TTTTCATA |
| G48S | CTGACCAGTATTCAGAAACTGGATATTCGTGT TATTGAAG | CTGAATACTGGTCAGAATATTACCACCACCAA TACCTTCC |

*Appendix 1—table 4 Continued on next page*

*Appendix 1—table 4 Continued*

| Mutation | Forward primer | Reverse primer |
|---|---|---|
| T79N | AATCGTAACTATACCAGCTATCCGGCAAAAATCCCGGATT | GGTATAGTTACGATTGCCATACTGAAATGCAACGCTCAGA |
| V127L | AAATTTCTGGGCAAAATCAAATATAACGGTGAAGGTTTCC | TTTGCCCAGAAATTTACCATCCTCGATGCTAATATCGCTT |
| G76D | CAGTATGACAATCGTACCTATACCAGCTATCCGGCAAAAA | ACGATTGTCATACTGAAATGCAACGCTCAGAATATCAAAG |
| G42A | ATTGGTGCTGGTAATATTCTGACCGGTATTCAGAAACTGG | ATTACCAGCACCAATACCTTCCATGCTAAAGGCATGACCA |
| H183R | ACCTGTCGCATGAAAACCATTTATCGCAGCAAAAAACCGG | TTTCATGCGACAGGTATAATGGGTGCTCTGGGTTTTATAG |
| K129R | GTGGGCAGAATCAAATATAACGGTGAAGGTTTCCCGGAAG | TTTGATTCTGCCCACAAATTTACCATCCTCGATGCTAATA |
| S155I | CCGGGCATCGAAAGCATGTATGTTAGTGATGGCACCCTGG | GCTTTCGATGCCCGGTTCCAGTTTGGTAACTTCTTTTTTC |
| E168D | GTTGGTGATGTTGTTCTGAGCTATAAAACCCAGAGCACCC | AACAACATCACCAACCAGGGTGCCATCACTAACATACATG |
| G163D | AGTGATGACACCCTGGTTGGTGAAGTTGTTCTGAGCTATA | CAGGGTGTCATCACTAACATACATGCTTTCGCTGCCCGGT |
| L197M | GAAAACATGCCGAAATTTCATTATGTTCATCACCGCCTGG | TTTCGGCATGTTTTCAACCGGTTTTTTGCTGCGATAAATG |
| O229Y | AAACCGTATTAAGAGCTCCAATCGCTTGGACCAGCTTTCC | CTCTTAATACGGTTTGGCAATTGCGGTTTCATGCTGCTCG |
| R206L | CATCACCTCCTGGAAAAAAAAAATTGTGGAAGAGGGCTATT | TTCCAGGAGGTGATGAACATAATGAAATTTCGGCAGGTTT |
| S106N | ACCCTGAACTTTGAAGATGGTGCCATTGTTAAAGTGGAAA | TTCAAAGTTCAGGGTACGTTCAAAGGTAAAACCTTCCGGA |
| S177N | ACCCAGAACACCCATTATACCTGTCACATGAAAACCATTT | ATGGGTGTTCTGGGTTTTATAGCTCAGAACAACTTCACCA |
| S155N | CCGGGCAACGAAAGCATGTATGTTAGTGATGGCACCCTGG | GCTTTCGTTGCCCGGTTCCAGTTTGGTAACTTCTTTTTTC |
| S172R | GTTCTGAGATATAAAACCCAGAGCACCCATTATACCTGTC | TTTATATCTCAGAACAACTTCACCAACCAGGGTGCCATCA |
| V148I | AAAGAAATTACCAAACTGGAACCGGGCAGCGAAAGCATGT | TTTGGTAATTTCTTTTTTCATAACCGGACCATCTTCCGGG |
| R206H | CATCACCACCTGGAAAAAAAAAATTGTGGAAGAGGGCTATT | TTCCAGGTGGTGATGAACATAATGAAATTTCGGCAGGTTT |
| H183R | ACCTGTCGCATGAAAACCATTTATCGCAGCAAAAAACCGG | TTTCATGCGACAGGTATAATGGGTGCTCTGGGTTTTATAG |

**Appendix 1—table 5.** Plasmids used in this study.

| Plasmid name | Description | Source | Addgene number |
|---|---|---|---|
| pAND | Source of backbone | Addgene #49377 | #49377 |
| pAND-MCS | MCS added, NdeI site removed from TetR | This study | #223514 |
| pDUP | *cogfp* under $P_{tac}$ | This study | #223515 |
| pDUP1 | *cogfp* inactive under $P_{tet}$, *cogfp* under $P_{tac}$ | This study | #223516 |
| pDUP2 | *cogfp* under $P_{tet}$, *cogfp* under $P_{tac}$ | This study | #223517 |

