## [Editor Report · eLife Assessment]

This **fundamental** study uses a creative experimental system to directly test Ohno's hypothesis, which describes how and why new genes might evolve by duplication of existing ones. In agreement with existing criticism of Ohno's original idea, the authors present **compelling** evidence that having two gene copies does not speed up the evolution of a new function as posited by Ohno, but instead leads to the rapid inactivation of one of the copies through the accumulation of mostly deleterious mutations. These findings will be of broad interest to evolutionary biologists and geneticists.

---

## [Referee Report · Reviewer #1 (Public review)]

The authors construct a pair of *E. coli* populations that differ by a single gene duplication in a selectable fluorescent protein. They then evolve the two populations under differing selective regimes to assess whether the end result of the selective process is a "better" phenotype when starting with duplicated copies. Importantly, their starting duplicated population is structured to avoid the duplication-amplification process often seen in bacterial artificial evolution experiments. They find that while duplication increases robustness and speed of adaptation, it does not result in more highly adapted final states, in contrast to Ohno's hypothesis.

Comments on revised version:

The authors have addressed my prior concerns, and I have no further comments on the manuscript.

---

## [Referee Report · Reviewer #2 (Public review)]

Summary:

Drawing from tools of synthetic biology, Mihajlovic et al. use a cleverly designed experimental system to dissect Ohno's hypothesis, which describes the evolution of functional novelty on the gene-level through the process of duplication & divergence.

Ohno's original idea posits that the redundancy gained from having two copies of the same gene allows one of them to freely evolve a new function. To directly test this, the authors make use of a fluorescent protein with two emission maxima, which allows to apply different selection regimes (e.g. selection for green AND blue, or, for green NOT blue). To achieve this feat without being distracted by more complex evolutionary dynamics caused by the frequent recombination between duplicates, the authors employ a well-controlled synthetic system to prevent recombination: Duplicates are placed on a plasmid as indirect repeats in a recombination-deficient strain of *E. coli*. The authors implement their directed evolution approach through in vitro mutagenesis and selection using fluorescent-activated cell sorting. Their in-depth analysis of evolved mutants in single-copy versus double-copy genotypes provides clear evidence for Ohno's postulate that redundant copies experience relaxed purifying selection. In contrast to Ohno's original postulate, however, the authors go on to show that this does not in fact lead to more rapid phenotypic evolution, but rather, the rapid inactivation of one of the copies.

Strengths:

This paper contributes with great experimental detail to an area where the literature predominantly leans on genomics data. Through the use of a carefully-designed, well-controlled synthetic system the authors are able to directly determine the phenotype & genotype of all individuals in their evolving populations and compare differences between genotypes with a single or double copy of coGFP. With it they find clear evidence for what critics of Ohno's original model have termed "Ohno's dilemma", the rapid non-functionalization by predominantly deleterious mutations.

Including an expressed but non-functional coGFP in (phenotypically) single copy genotypes provides an especially thoughtful control that allows determining a baseline dN/dS ratio in the absence of selection. All in all the study is an exciting example of how the clever use of synthetic biology can lead to new insights.

Weaknesses:

In the revised version of the paper, the authors now discuss one potential weakness of their study, which is tied to its biggest strength (as often in experimental biology there is a trade-off between 'resolution' and 'realism').

The experimental set-up leaves out an important component of the evolutionary process in order to disentangle dosage effects from other effects that carrying two copies might have on their evolution. Specifically, by employing a recombination-deficient strain and constructing their duplicates as inverted repeats their experimental design completely abolishes recombination between the two copies. This was pointed out in my first review to be problematic for two reasons:

(i) In nature, new duplicates do not arise as inverted, but rather as direct (tandem) repeats and - as the authors correctly point out - these are very unstable, due to the fact that repeated DNA is prone to recA-dependent homologous recombination (which arise orders of magnitude more frequently than point mutations).

(ii) This instability often leads to further amplification of the duplicates under dosage selection both in the lab and in the wild (e.g. Andersson & Hughes, Annu. Rev. Genet. 2009), and would presumably also be an outcome under the current experimental set-up if it was not prevented from happening?

In their revised version, the authors now address this point and with much clarity explain why their experimental system is so powerful to study the fate of a gene duplicate, not despite lacking recombination, but *because* it lacks recombination.

---

## [Author Response]

The following is the authors’ response to the original reviews.

**eLife assessment**
This fundamental study uses a creative experimental system to directly test Ohno's hypothesis, which describes how and why new genes might evolve by duplication of existing ones. In agreement with existing criticism of Ohno's original idea, the authors present compelling evidence that having two gene copies does not speed up the evolution of a new function as posited by Ohno, but instead leads to the rapid inactivation of one of the copies through the accumulation of mostly deleterious mutations. These findings will be of broad interest to evolutionary biologists and geneticists.

We thank the editors and the reviewers for their positive feedback concerning our experimental system and for the constructive feedback on how to further improve the manuscript. We have now addressed the reviewer’s comments in a revised version.

**Reviewer #1 (Public Review):**
Overview:The authors construct a pair of *E. coli* populations that differ by a single gene duplication in a selectable fluorescent protein. They then evolve the two populations under differing selective regimes to assess whether the end result of the selective process is a "better" phenotype when starting with duplicated copies. Importantly, their starting duplicated population is structured to avoid the duplication- amplification process often seen in bacterial artificial evolution experiments. They find that while duplication increases robustness and speed of adaptation, it does not result in more highly adapted final states, in contrast to Ohno's hypothesis.Major comments:This is an excellent study with a very elegant experimental setup that allows a precise examination of the role of duplication in functional evolution, exclusive of other potential mechanisms. My main concern is to clarify some of the arguments relating to Ohno's hypothesis.I think my main confusion on first reading the manuscript was in the precise definition of Ohno's hypothesis. I think this confusion was mine and not the authors, but it is likely common and could be addressed.Most evolutionary biologists think of gene duplication as making neofunctionalization "easier" by providing functional redundancy and a larger mutational target, such that the evolutionary process of neofunctionalization is faster (as the authors observed). In this framework, the final evolved state might not differ when selection is applied to duplicated copies or a single-copy gene. Ohno's hypothesis, by contrast, argues that there generally exist adaptive conflicts between the ancestral function and the "desired" novel function, such that strong selection on a single-copy gene cannot produce the evolutionary optima that selection on two copies would. This idea is hinted at in the quotation from Ohno in paragraph 2 of the introduction. However, the sentences that follow I don't think reinforce this concept well enough and lead to some confusion.With that definition in mind, I agree with the authors' conclusion that these data do not support Ohno's hypothesis. My quibble would be that what is actually shown here is that adaptive conflict in function is not universal: there are cases where a single gene can be optimized for multiple functions just as well as duplicated copies. I do not think the authors have, however, refuted the possibility that such adaptive conflicts are nonetheless a significant barrier to evolutionary innovation in the absence of gene duplication generally. Perhaps just a sentence or two to this effect might be appropriate.

We fully agree with the reviewer that trade-offs might play an important role in the evolution of single copy and of duplicated genes, depending on the gene and on the selection regime. And while trade-offs are not likely to play a big role in the selection regime we discuss in detail in the main text (evolution towards more green), they probably are important for at least one our selection regimes. In fact, we so state in the following passage of the discussion. In addition, we have now added a sentence that acknowledges the importance of trade-offs for evolution in the absence of gene duplication:

“A single gene encoding such a protein suffers from an adaptive conflict between the two activities. Gene duplication may provide an escape from this adaptive conflict, because each duplicate may specialize on one activity14, 15. For coGFP, a trade-off likely exists for fluorescence in these two colors, because improvement of green fluorescence entails a loss of blue fluorescence during evolution (Figure S8 and Figure S16). We therefore expected that during selection for both green and blue fluorescence, one *cogfp* copy in double-copy populations would “specialize” on green fluorescence whereas the other copy would specialize on blue fluorescence. However, when we analyzed individual population members with two active gene copies we could not find any such specialization (Figure S21). Moreover, the identified key mutations at positions 147 and 162 have a very low frequency (<1%) in these populations (Figure S15). Future experiments with different selection strategies might reveal the reasons for this observation and the conditions under which such a specialization can occur.“

I also think the authors need to clarify their approach to normalizing fluorescence between the two populations to control for the higher relative protein expression of the population with a duplicated gene. Since each population was independently selected with the highest fluorescing 60% (or less) of the cells selected, I think this normalization is appropriate. Of course, if the two populations were to compete against each other, this dosage advantage of the duplicates would itself be a selective benefit. Even as it is, the dosage advantage should be a source of purifying selection on the duplication, and perhaps this should be noted.

The reviewer is correct. To be able to follow the evolutionary trajectories of the different constructs, the populations were treated separately. The gates were adjusted for each library separately to select for the top 60, 1 or 0.01% of cells and the gates for the double-copy populations were set to slightly higher fluorescence, reflected in the higher fluorescence of these populations in Figure 3A. Indeed, if individuals in these populations were to compete against each other, the double-copy populations would have a benefit due to the dosage advantage. However, as we already pointed out in the manuscript, we did not see any additional advantage beyond the increased gene dosage provided by the second copy (Figure 3B). To discuss this issue in more detail, we have now added the following text to the discussion:

“It is worth noting that we evolved each of our single- and double-copy populations separately and in parallel to follow their individual evolutionary trajectories. In a natural population, individuals with one or two copies might occur in the same population and compete against each other. In this situation any dosage advantage of a duplicate gene would itself entail selective benefit. Our approach allowed us to find out if gene duplication facilitates phenotypic evolution beyond any such gene dosage effect. At least for the specific genes, selection pressures, and mutation rates we used, the data suggest that it does not.”

Finally, I am slightly curious about the nature of the adaptations that are evolving. The authors primarily discuss a few amino-acid changing mutations that seem to fix early in the experiment. Looking at Figure 3, it however, appears that the populations are still evolving late in the experiment, and so presumably other changes are occurring later on. Do the authors believe that perhaps expression changes to increase protein levels are driving these later changes?

Figure S15 shows that some mutations are indeed still increasing in frequency during late evolutionary rounds, in particular S2L, V141L and V205L. We have measured the emission spectra of these mutants (Figure S16), and these mutations increase fluorescence both in green and blue. It is therefore likely that these mutations, similar to L98M, increase protein expression, solubility, or thermal stability, as suggested by the reviewer. We now clarify this matter in a new passage of the results:

“Like L98M, the additional mutations S2I, V141I and V25L also occurred in all selection regimes, but they reached lower frequencies than L98M during the 5 generations of the experiment. We hypothesized that mutations observed in all selection regimes do not derive their benefit from increasing the intensity of any one fluorescent color. Instead, they may increase protein expression, solubility, or thermal stability.”

**Reviewer #2 (Public Review):**
Summary:Drawing from tools of synthetic biology, Mihajlovic et al. use a cleverly designed experimental system to dissect Ohno's hypothesis, which describes the evolution of functional novelty on the gene-level through the process of duplication & divergence.Ohno's original idea posits that the redundancy gained from having two copies of the same gene allows one of them to freely evolve a new function. To directly test this, the authors make use of a fluorescent protein with two emission maxima, which allows them to apply different selection regimes (e.g. selection for green AND blue, or, for green NOT blue). To achieve this feat without being distracted by more complex evolutionary dynamics caused by the frequent recombination between duplicates, the authors employ a well-controlled synthetic system to prevent recombination: Duplicates are placed on a plasmid as indirect repeats in a recombination-deficient strain of *E. coli*. The authors implement their directed evolution approach through in vitro mutagenesis and selection using fluorescent-activated cell sorting. Their in-depth analysis of evolved mutants in single-copy versus double-copy genotypes provides clear evidence for Ohno's postulate that redundant copies experience relaxed purifying selection. In contrast to Ohno's original postulate, however, the authors go on to show that this does not in fact lead to more rapid phenotypic evolution, but rather, the rapid inactivation of one of the copies.Strengths:This paper contributes with great experimental detail to an area where the literature predominantly leans on genomics data. Through the use of a carefully designed, well-controlled synthetic system the authors are able to directly determine the phenotype & genotype of all individuals in their evolving populations and compare differences between genotypes with a single or double copy of coGFP. With it they find clear evidence for what critics of Ohno's original model have termed "Ohno's dilemma", the rapid non- functionalization by predominantly deleterious mutations.Including an expressed but non-functional coGFP in (phenotypically) single copy genotypes provides an especially thoughtful control that allows determining a baseline dN/dS ratio in the absence of selection. All in all the study is an exciting example of how the clever use of synthetic biology can lead to new insights.Weaknesses:The major weakness of the study is tied to its biggest strength (as often in experimental biology there is a trade-off between 'resolution' and 'realism').The paper ignores an important component of the evolutionary process in favour of an in-depth characterization of how two vs one copy evolve. Specifically, by employing a recombination-deficient strain and constructing their duplicates as inverted repeats their experimental design completely abolishes recombination between the two copies.This is problematic for two reasons:i) In nature, new duplicates do not arise as inverted, but rather as direct (tandem) repeats and - as the authors correctly point out - these are very unstable, due to the fact that repeated DNA is prone to recA- dependent homologous recombination (which arise orders of magnitude more frequently than point mutations).ii) This instability often leads to further amplification of the duplicates under dosage selection both in the lab and in the wild (e.g. Andersson & Hughes, Annu. Rev. Genet. 2009), and would presumably also be an outcome under the current experimental set-up if it was not prevented from happening?So in sum, recombination between duplicate genes is not merely a nuisance in experiments, but occurring at extremely high frequencies in nature (such that the authors needed to devise a clever engineering solution to abolish it), and is often observed in evolving populations, be it in the laboratory or the wild.The manuscript sells controlling of copy number as a strength. And clearly, without it, the same insights could not be gained. However, if the basis for the very process of what Ohno's model describes is prevented from happening for the process to be studied, then, for reasons of clarity and context this needs pointing out, especially, to readers less familiar with the principles of molecular evolution.Connected to this, there are several places in the introduction and the discussion where I feel that the existing literature, in particular models put forward since Ohno that invoke dosage selection (such as IAD) end up being slightly misrepresented.My point is best exemplified in line 1 of Discussion: "To test Ohno's hypothesis and to distinguish its predictions from those of competing hypotheses, it is necessary to maintain a constant and stable copy number of duplicated genes during experimental evolution."

We understand the reviewer’s position and fully agree that we needed to clarify better what our experiments aimed to achieve. To this end, we rewrote the beginning of the discussion to read:

“Our aim was to study whether gene duplication can affect mutational robustness and phenotypic evolution beyond any effect of increased gene dosage provided by multiple gene copies. To this end, we needed to maintain a constant and stable copy number of duplicated genes during experimental evolution.”

I think this statement is simply not true and might be misleading. To take the exaggerated position of a devil's advocate, the goal of evolutionary biology should be to find out how evolution actually proceeds in nature most of the time, rather than creating laboratory systems that manage to recapitulate influential ideas.

On this point, we respectfully disagree. To ask questions like ours, laboratory experiments that are highly controlled albeit possibly “unnatural” can be essential. And we would argue that our experiments do not merely aim to “recapitulate” an influential idea but to validate it and potentially refute it, as we did for our study system. Validating theory is an essential aspect of experimental science. Textbooks in biology and beyond are rife with examples.

While fixing copy number may be a necessary step to understand how one copy evolves if a second one is present, it seems that if Ohno's hypothesis only works out in recA-deficient bacterial strains and on engineered inverted repeats, that Ohno might have missed one crucial aspect of how paralogs evolve. The mentioned competing hypotheses have been put forward to (a) address Ohno's dilemma (which the present study beautifully demonstrates exists under their experimental conditions) and (b) to reflect a commonly observed evolutionary process in bacteria (dosage gain in response to selection, e.g. a classic way of gaining antibiotic resistance). Fixing the copy number allowed the authors to show which predictions of Ohno's model hold up and which don't (under these specific conditions). But they do so without even preventing the processes described by alternative models from happening, so the experimental system is hardly appropriate to distinguish between Ohno & alternatives. Therefore, I think it could be made clearer that the experimental system is great to look at certain aspects Ohno's hypothesis in detail, but it can only inform us about a universe without recombination.(1) Citing the works by ref 8, 26, 27 to merely state that "in some copies were gained and some were lost (ref 6, ref 25)" makes it seem as if fixing at 2 copies is some sort of sensible average. Yet ref 6 (Dhar et al.) specifically states that dosage is the most important response. Moreover, in this study gene copies are lost, but plasmid copies are gained instead. In Holloway et al. 2007 (ref 25), the 2 copies resided on different plasmids, so entirely different underlying molecular genetics might be at work (high cost of plasmid maintenance, and competitive binding on both proteins onto the respective (off)-target, where either way selection favored a single copy, so a different situation altogether). In both cited studies, fixing the copy would have prohibited learning something about the process of duplication & divergence.Hence this statement seems to distract the readers from the main message, which seems that preventing recombination experimentally allows to follow the divergence of each copy and studying a response that does not involve dosage-increase.(2) "These studies highlighted the importance of gene duplication in providing fast adaptation under changing environmental conditions but they focused on the importance of gene dosage." I think this constructs a false dichotomy. Instead, these studies pointed out that dosage (and with it, selection for dosage) is an important part of the equation that might have been missed by Ohno.

Your points are well taken. To clarify the insights from previous experiments and the aims of our experiments we rewrote this passage in question as follows.

“These studies underline the importance of gene duplication in providing fast adaptation under changing environmental conditions. In some studies one copy was lost6, 25, while in others, additional copies were gained8, 26, 27. Together these studies highlight that gene dosage and selection for dosage can play an important role during the evolution of duplicated genes6, 8, 25-28.

These studies also raise the question whether gene duplication can provide an advantage beyond its effects on gene dosage. To find out it is necessary to study the evolution of gene duplicates while keeping the copy number of the duplicated gene exactly at two. This is challenging because gene duplication causes recombinational instability and high variability in copy number. No previous experimental studies were designed to control copy number. Here, we present an experimental system that allowed us to keep the copy number fixed at one or two genes, and to follow the evolution of each gene copy in the absence of any dosage increase.”

(3) "Such models are also easier to test experimentally, because they do not require precise control of gene copy number. The necessary tests can even benefit from massive gene amplifications8. Although Ohno's hypothesis is more difficult to test experimentally (...)" - again, I feel the wording is slightly misleading. The point is not that IAD is easier to test and Ohno's model is harder to test in laboratory experiments, rather, experiments (and some more limited observations of naturally evolving populations) seem to suggest that in reality evolution proceeds (more often?) according to IAD rather than Ohno's neofunctionalization hypothesis. However, as the authors point out, it will be exciting to see their clever experimental system used to test other genes and conditions to get a more comprehensive understanding of what gene- and selection- parameter values would overcome Ohno's dilemma.

We agree and in response rewrote the paragraph in question to read:

“The challenge that a duplicated gene copy must remain free of frequent deleterious mutations long enough to acquire beneficial mutations that provide a new selectable phenotype is known as Ohno’s dilemma13. Our experiments confirm that this challenge is highly relevant for post-duplication evolution. Other models such as the innovation-amplification-divergence (IAD) model8, 13 postulate that this dilemma can be resolved through an increase in gene dosage that allows latent pre-duplication phenotypes to come under the influence of selection. To distinguish between the effects of gene dosage and other benefits of gene duplication, we prevented recombination and gene amplification to prevent copy number increases beyond two copies. We are aware that our experimental design does not reflect how evolution may occur in the wild. However, this design allowed us to study evolutionary forces separately that are otherwise difficult to disentangle. “

Finally, we also made two changes in the abstract (highlighted in red) to take your feedback into account.

**Reviewer #2 (Recommendations For The Authors):**
The paper is very well written, with a lot of emphasis put on explaining every step and every finding. It was a joy to read.

Thanks!

Full stop missing in line 5 of abstract.

Corrected.